# Autophagy acts through TRAF3 and RELB to regulate gene expression via antagonism of SMAD proteins

Alice C. Newman [1,2], Alain J. Kemp[1], Yvette Drabsch[1], Christian Behrends [3,4] & Simon Wilkinson [1]

Macroautophagy can regulate cell signalling and tumorigenesis via elusive molecular mechanisms. We establish a RAS mutant cancer cell model where the autophagy gene *ATG5* is dispensable in A549 cells in vitro, yet promotes tumorigenesis in mice. ATG5 represses transcriptional activation by the TGFβ-SMAD gene regulatory pathway. However, autophagy does not terminate cytosolic signal transduction by TGFβ. Instead, we use proteomics to identify selective degradation of the signalling scaffold TRAF3. TRAF3 autophagy is driven by RAS and results in activation of the NF-κB family member RELB. We show that RELB represses TGFβ target promoters independently of DNA binding at NF-κB recognition sequences, instead binding with SMAD family member(s) at SMAD-response elements. Thus, autophagy antagonises TGFβ gene expression. Finally, autophagy-deficient A549 cells regain tumorigenicity upon SMAD4 knockdown. Thus, at least in this setting, a physiologic function for autophagic regulation of gene expression is tumour growth.

[1] Edinburgh Cancer Research UK Centre, MRC Institute of Genetics and Molecular Medicine, University of Edinburgh, Edinburgh EH4 2XR, UK. [2] Wolfson Wohl Cancer Research Centre, Institute of Cancer Sciences, University of Glasgow, Garscube Estate, Glasgow G61 1QH, UK. [3] Munich Cluster for Systems Neurology (SyNergy), Ludwig-Maximilians-University, Feodor-Lynen-Str. 17, 81377 Munich, Germany. [4] Institute of Biochemistry II, Goethe University School of Medicine, Theodor Stern Kai 7, Frankfurt am Main 60590, Germany. Alain J. Kemp and Yvette Drabsch contributed equally to this work. Correspondence and requests for materials should be addressed to S.W. (email: simon.wilkinson@igmm.ed.ac.uk)

Macroautophagy (hereafter autophagy) is a major cytosolic degradative pathway that participates in cellular metabolism, homoeostasis and anti-microbial defence[1]. Upstream stress signals converge on proteins involved in biogenesis of a double-membraned vesicle known as the autophagosome[2, 3]. This core autophagy machinery includes ATG5, which predominantly exists in a protein–protein conjugate with ATG12 (ATG5-12), and other key players such as the FIP200/ULK1 complex. These proteins act upstream of the recruitment of ATG8-family ubiquitin-like proteins, such as LC3B, to nascent autophagic membranes, via lipidation of their C-terminal glycine residues with phosphatidylethanolamine. Fully formed, enclosed autophagosomes sequester cytosolic cargo that is in turn degraded upon autophagosomal–lysosomal fusion.

Autophagic cargo can comprise general cytosol. However, autophagy pathways may also select specific cargoes for degradation, for example damaged mitochondria, bacteria or protein aggregates[4]. Notably, termination of cytosolic signalling events by selective autophagy (signalphagy) is emerging as an important modulator of cell fate, although this has been less widely analysed[5–10]. Selective autophagy is facilitated by bifunctional 'cargo receptors' that bind both to ATG8-family proteins, and, directly or indirectly, to selected ubiquitinated cargoes[4]. The prototypical cargo receptor is p62 (SQSTM1)[11, 12]. However, other, less well-characterised cargo receptors also participate, including nuclear dot protein 52 kDa (NDP52), which was identified first as a mediator of bacterial autophagy and latterly as a component of the mitochondrial autophagy apparatus[13–16].

A rich, yet complex, scenario for unravelling signalling functions of selective autophagy is tumorigenesis. RAS small GTPases are oncogenically activated in numerous cancers and generally drive elevated autophagy activity in order to support tumorigenesis[17–23], with some notable exceptions[24]. Altered metabolism and mitophagy may have a role here[17, 20]. However, other molecular mechanisms remain to be identified. Hypothetically, these could encompass signalphagy events that would participate in signalling cross-talk downstream of RAS with other tumour-relevant pathways and consequently mediate reprogramming of gene expression. Indeed, some recent studies illustrate the potential for gene regulation by autophagy, such as inhibition of inflammatory gene expression via degradation of TBK1 and its substrate, the transcription factor IRF3[9, 10], or senescence-associated degradation of the transcription factor GATA4[25]. Nonetheless, the prevalence of signalphagy-mediated transcriptional regulation is largely unexplored. We recently proposed that non-canonical (alternative) NF-κB signalling, involving the RELB transcription factor, may be dependent upon ATG5, presumably via an as-yet-unidentified selective autophagy pathway[26]. However, the mechanism and significance of this is unclear.

An important signalling molecule that regulates gene expression is transforming growth factor β (TGFβ)[27]. TGFβ ligates receptor serine–threonine kinases, ultimately resulting in cytosolic phosphorylation of selected transcription factors of the SMAD family, such as SMAD2 and SMAD3. Contingent upon this, heteromeric SMAD assemblies, such as SMAD 2/2/4, SMAD 3/3/4 and, possibly, SMAD2/3/4 complexes, translocate to the nucleus and bind SMAD-response element (SREs) at proximal promoters to drive transcription[27]. The TGFβ transcriptome exerts pleiotropic effects on tumour biology[28, 29]. On one hand, it can inhibit cell cycle progression and promote apoptosis. On the other hand, TGFβ-driven transcriptional changes also underpin epithelial–mesenchymal transition (EMT) and enhanced metastatic abilities of cancer cells. The latter occurs particularly during cancer progression when resistance or insensitivity to the anti-proliferative effects of TGFβ are evident. Such insensitivity may be acquired during the evolution of a tumour. Indeed, RAS mutant cancer cells commonly exhibit decreased sensitivity to the anti-tumorigenic effects of the TGFβ ligand[30]. In certain settings, such as some pancreatic cancers, this may occur by mutation, for example deletion of SMAD4[31]. However, resistance of RAS-driven cancer cells to anti-tumorigenic effects of TGFβ may occur via alternate, unknown mechanism(s) in other settings.

Here we show that autophagy is required for tumour formation in mice by RAS-mutant cancer cells. We identify transcriptional reprogramming via the SMAD proteins when autophagy is inhibited. We discover that a SMAD–RELB complex ordinarily represses transcription at TGFβ target genes. This is independent of the conventional DNA-recognition activity of RELB but requires indirect recruitment of RELB to genes via SMAD(s). Activation of RELB, and consequent antagonism of TGFβ, occurs specifically by RAS-mediated engagement of autophagy. This autophagy facilitates NDP52-mediated degradation of the signal terminator for the alternative NF-κB pathway, TRAF3. Finally, cross-talk with TGFβ via autophagy/RELB is required for the promotion of tumorigenesis in mice by A549 lung cancer cells.

## Results

**Autophagy promotes tumorigenesis.** To investigate the effect of stable autophagy inhibition in RAS-mutated human cancer cells, we used CRISPR/Cas9 to ablate *ATG5* expression in A549 lung adenocarcinoma cells (Fig. 1a, b). Inhibition of autophagy was confirmed by accumulation of unmodified LC3B-I protein at the expense of lipidated LC3B-II (Fig. 1a, ΔATG5 vs. wild-type, WT, controls). Surprisingly, ΔATG5 cells had no differences in growth kinetics compared to controls (Fig. 1c). This is in contrast to the reported anti-proliferative effects of acute autophagy inhibition in numerous RAS-mutated human cell types[17, 19, 26]. Indeed, RNA interference (RNAi) against *ATG5* and *FIP200* inhibited proliferation of the parental A549 cell population (Supplementary Fig. 1a, b), but notably did not kill cells (Supplementary Movies 1–3). Thus, autophagy is not required for the long-term proliferative potential of cells in vitro.

We next investigated in vivo tumorigenicity, using a subcutaneous xenograft model. To ensure that differences between WT and ΔATG5 cells were attributable to autophagy, we stably expressed GFP-ATG5 in one clone of ΔATG5 cells, rescuing LC3B lipidation and p62 regulation (Fig. 1d). Using these cell lines (Fig. 1e) or an alternate pair of wild-type and *ATG5*-deleted clones (Fig. 1f), we observed that loss of *ATG5* markedly diminished tumour growth kinetics in vivo. Consistent with the in vitro analyses, impairment of autophagy was detected in ΔATG5 tumours, as shown by a loss of LC3B puncta by immunohistochemistry (Fig. 1g).

The above data show that sustained and complete loss of autophagy does not necessarily impede RAS-mutant cancer cell proliferation in vitro. However, autophagy can be a contributor to physiological tumour growth in this same cell type.

**Autophagy represses TGFβ-driven gene expression.** We next sought to identify mechanism(s) by which autophagy was promoting tumorigenesis. We performed global gene expression analysis of A549 cells after RNAi against core autophagy genes *ATG5* or *ULK1* (Supplementary Data set 1). These siRNA reagents have previously been validated in these cells[26]. We aimed to extract differentially expressed, ontologically coherent transcript sets that would illuminate transcriptional reprogramming events regulating tumorigenesis. Accordingly, computational gene set enrichment analysis (GSEA) was performed[32]. Changes in transcript levels upon autophagy inhibition were compared with known oncogenic and tumour suppressive pathway readouts. Surprisingly, the highest scoring correlation was with transcripts

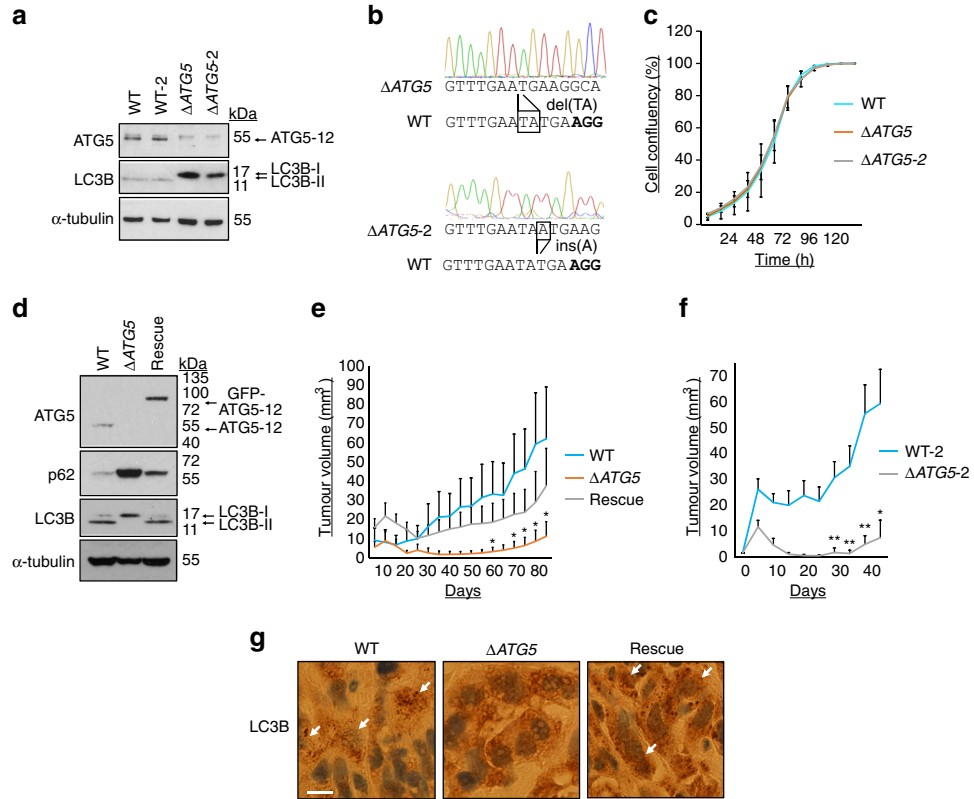

**Fig. 1** Autophagy promotes tumorigenesis in vivo. **a** A549 cells deficient in ATG5 protein expression were generated by CRISPR/Cas9-mediated genome editing. Each of the two wild-type control clones (WT and WT-2) and ΔATG5 clones (ΔATG5 and ΔATG5-2) were immunoblotted for the indicated proteins. **b** Genomic DNA from A549 ΔATG5 and ΔATG5-2 cells was PCR amplified. The amplicon encompassed the sequence to which Cas9 had been targeted (PAM in *bold*) and different indels were identified in both clones via Sanger sequencing. **c** A549 WT and ΔATG5 cells were plated at low confluency for time-lapse phase-contrast videomicroscopy using an Incucyte microscope and cell proliferation was monitored by automated confluency analysis at set intervals post plating (means, $n = 9$ wells, ±S.D.). **d** A pooled derivative of ΔATG5 cells was generated by stable transduction with *GFP-ATG5* retrovirus (rescue). The indicated cell lines were immunoblotted as shown. **e** WT, ΔATG5 and rescue cells were subcutaneously injected into immunocompromised mice and tumour volume was measured longitudinally (means, $n = 10$ flanks, ±S.E.M., *$P < 0.05$ vs. WT, two-tailed $t$-test). **f** WT-2 and ΔATG5-2 cells were compared for tumour growth after subcutaneous injection into immunocompromised mice (means, $n = 12$ flanks, ±S.E.M., *$P < 0.05$ or **$P < 0.01$ vs. WT-2, two-tailed $t$-test). **g** At the end of tumour growth in **e**, control tumours and sufficiently large ΔATG5 tumours were fixed and stained for LC3B via immunohistochemistry (DAB stain, *arrows* indicate regions of LC3B puncta, *scale bar* = 10 μm). Representative images are shown here. Uncropped blots are available in Supplementary Fig. 10

that are upregulated by the growth factor TGFβ (Fig. 2a). Indeed, two-fifths of the 50 most autophagy-repressed transcripts were identified by manual curation as either direct or indirect targets of transcriptional upregulation by TGFβ, referred to collectively hereafter as 'TGFβ-driven genes' (Fig. 2b, consult Supplementary Table 1 for a detailed justification of gene classification).

We selected a subset of the transcripts identified in Fig. 2b for further analyses, in order to elucidate the molecular mechanisms underlying the apparent antagonism of TGFβ function by autophagy. Firstly, quantitative real-time PCR (qRT-PCR) after *ATG5* RNAi confirmed repression of TGFβ-driven genes by autophagy, either under basal growth conditions or upon stimulation by exogenous TGFβ ligand (Fig. 2c). *ULK1* RNAi also enhanced gene expression to a comparable extent (Fig. 2d). The persistent autophagy defect in non-tumorigenic A549 ΔATG5 cells was also associated with upregulation of such transcripts (Fig. 2e). Repression of TGFβ-driven genes by autophagy was also evident upon analysis of orthologous transcripts, basally or after TGFβ treatment, in RAS-transformed mouse embryonic fibroblasts (MEFs) where *Atg5* was deleted (Fig. 2f, MEF KRAS-V12 cells, congenic WT or *Atg5*−/−). Thus, the apparent TGFβ-inhibitory function of autophagy is manifest in different cell lineages and across species.

We inferred from the above data that autophagy antagonises TGFβ signalling, at least at the level of target gene output. Indeed, we confirmed that the basal levels of known TGFβ-driven transcripts in our data were dependent upon tonic TGFβ-dependent signal transduction, using an ALK2/4/5 receptor serine–threonine kinase inhibitor (ALKi) (Supplementary Fig. 2b). The cytosolic signalling events in the TGFβ pathway can be read out in phosphorylation of SMAD2 and SMAD3. However, no increases in basal or exogenous TGFβ-stimulated phospho-SMAD species were detected in A549 ΔATG5 or MEF KRAS-V12 *Atg5*−/− cells (Fig. 2g, h). These data raise the question of a non-cytosolic site of convergence between molecular events downstream of autophagy and TGFβ.

**RAS drives RELB signalling via autophagy of TRAF3.** We hypothesised that autophagy might selectively target cytosolic proteins with hitherto unknown functions in regulating the nuclear output of TGFβ signalling. To identify such proteins, we performed mass spectrometric screening for interactors of known cargo receptors. Of note, putative NDP52 interactors included a number of signal transduction mediators (Fig. 3a, Supplementary Data set 2). In particular, we identified tumour necrosis factor receptor-associated factor 3 (TRAF3), a cytosolic scaffold that

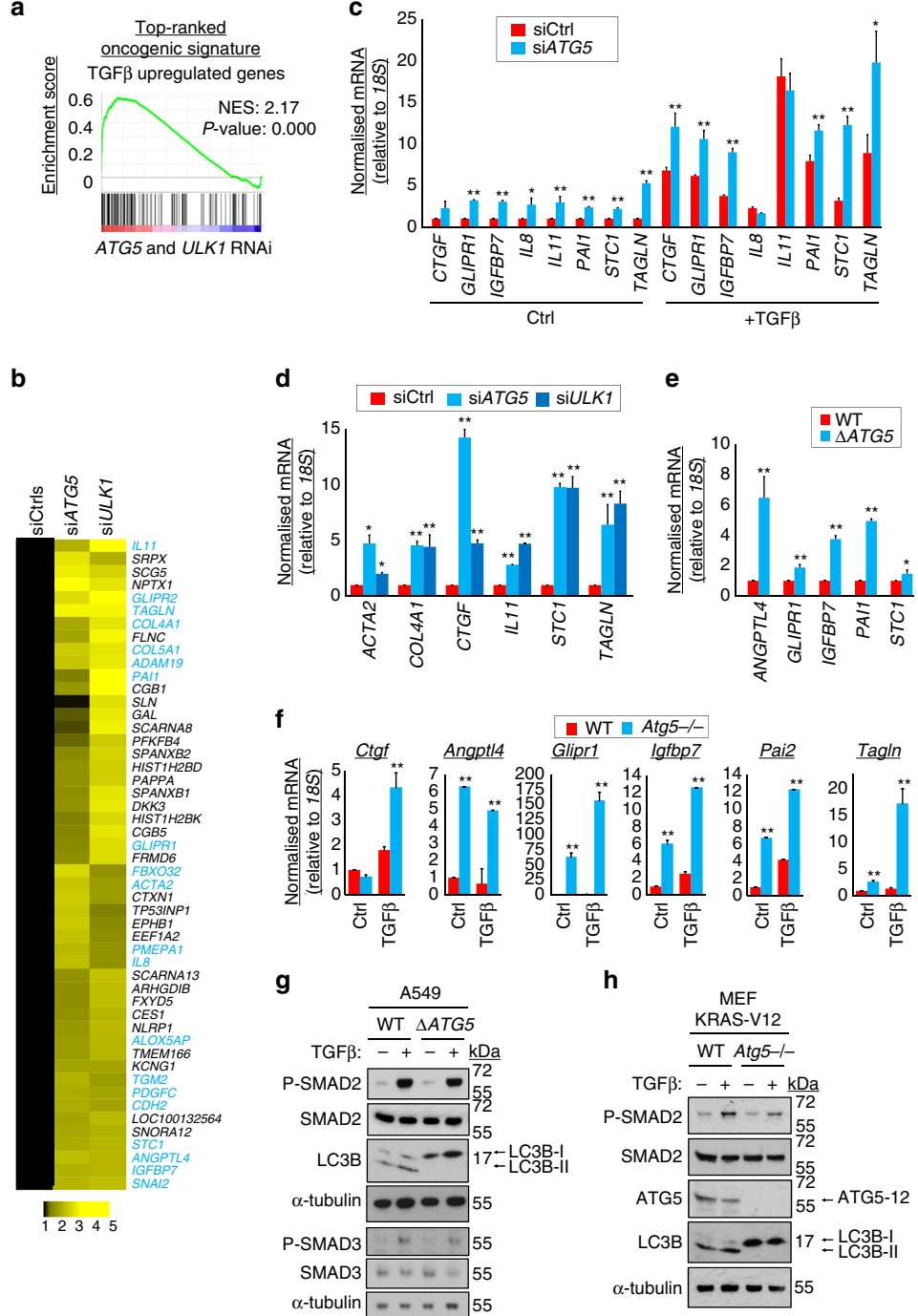

**Fig. 2** Autophagy suppresses TGFβ target gene expression. **a** siRNA oligonucleotides targeting *ATG5* and *ULK1,* or two sequence-unrelated non-targeting controls, were transfected in A549 cells in each of the three independent replicates. Expression profiles were obtained from an Illumina bead array, summarised in Supplementary Data set 1. Gene set enrichment analysis was performed, comparing genes differentially expressed in the autophagy inhibited cells with known oncogene- and tumour suppressor-driven expression profiles. The comparison yielding the greatest normalised enrichment score (NES), 'TGFβ upregulated genes', is shown here. **b** Fold-change heat map showing the top 50 upregulated genes after *ATG5* and *ULK1* RNAi. Mean si*ATG5* or si*ULK1* siRNA values are normalised to the averaged means of the two non-targeting controls (siCtrls). *Blue text* indicates genes that are known to be transcriptionally upregulated by TGFβ (Supplementary Table 1). **c–f** qRT-PCR was performed for the indicated transcripts after the following treatment regimens. **c** A549 cells were transfected with siCtrl or si*ATG5* for 72 h. Cells were either left untreated (Ctrl) or treated with 5 ng/ml TGFβ1 for the final 16 h of transfection. **d** A549 cells were transfected with siCtrl, si*ATG5* or si*ULK1* for 72 h. **e** Exponentially dividing A549 WT and Δ*ATG5* cells were compared. **f** Congenic wild-type (*WT*) or *Atg5* null (*Atg5−/−*) MEF KRAS-V12 cells were left untreated (Ctrl) or treated with 5 ng/ml TGFβ1 for 16 h (means, $n = 3$, ±S.D., *$P < 0.05$ or **$P < 0.01$ vs. siCtrl or similarly treated WT cells, two-tailed *t*-test). **g** A549 WT and Δ*ATG5* cells and **h** WT and *Atg5−/−* MEF KRAS-V12 cells, with (+) or without (−) prior treatment with 5 ng/ml TGFβ1 for 16 h, were lysed and immunoblotted (P-SMAD2 = phospho-S465/467 SMAD2, P-SMAD3 = phospho-S423/425 SMAD3). Uncropped blots are available in Supplementary Fig. 10

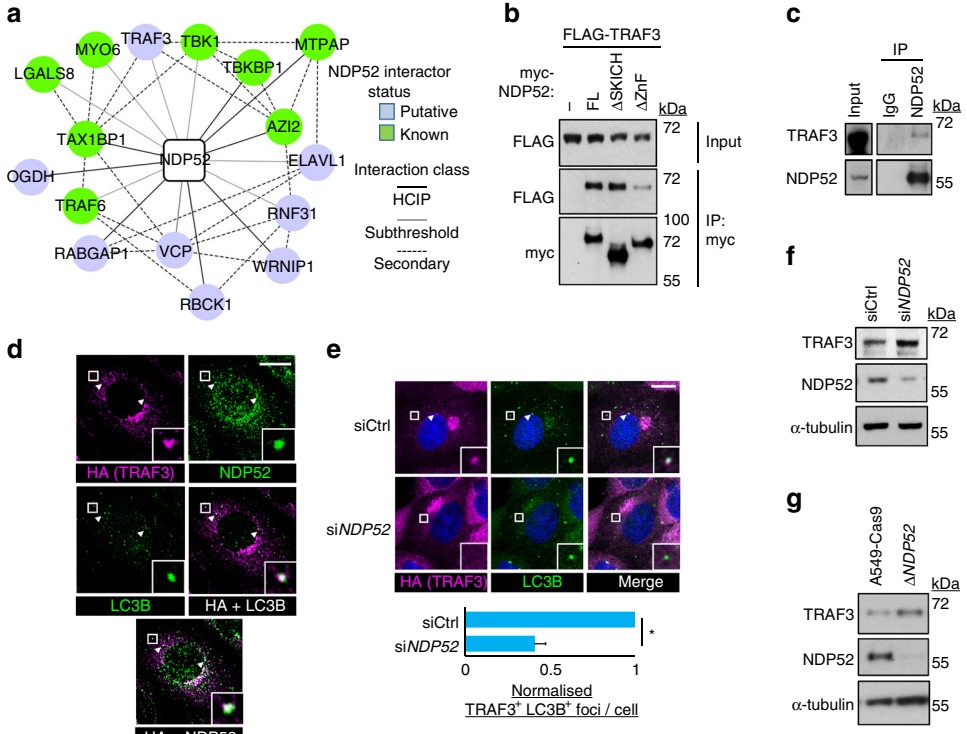

**Fig. 3** Autophagic degradation of TRAF3 via the cargo receptor protein NDP52. **a** FLAG-HA-NDP52 was used as bait for co-immunoprecipitation LC-MS/MS from A549 cells. Hits were identified using CompPASS thresholding for high-confidence interacting proteins (HCIPs) and by inclusion of immediate subthreshold interactors on the proviso of known secondary interaction with at least two HCIPs (BioGRID). Known interactors of NDP52 were equally distributed among HCIP and subthreshold hits. **b** HEK293T cells were co-transfected with FLAG-TRAF3 and either empty vector (−) or myc-tagged forms of NDP52 (full-length, FL, or deletants of the N-terminal SKICH domain, ΔSKICH, or C-terminal Zinc Fingers, ΔZnF). Anti-myc immunoprecipitation (IP) was performed followed by immunoblotting. **c** A549 lysates were immunoprecipitated with anti-NDP52 antibody-conjugated beads or control rabbit IgG beads and immunoblotted. **d** A549 cells stably expressing FLAG-HA-TRAF3 were stained for indicated epitopes and analysed by confocal microscopy (*scale bar* = 10 μm). LC3B and NDP52 are both false-coloured *green* and are overlaid separately with *magenta* HA (TRAF3) in merge panels. *Arrowheads* indicate co-localising foci. *Boxes* correspond to zoomed *insets*. **e** A549 cells expressing FLAG-HA-TRAF3 were transfected with the indicated siRNA for 72 h and then cells were stained for confocal microscopy (*scale bar* = 10 μm). *Arrowheads* indicate dual foci of TRAF3 and LC3B, quantified on a per cell basis (means, $n = 3$, ±S.E.M., *$P < 0.05$, two-tailed *t*-test). **f** A549 cells were transfected with siCtrl or si*NDP52* for 72 h and immunoblotted as shown. **g** A549-Cas9 control cells or pooled Δ*NDP52* counterparts were immunoblotted as shown. Uncropped blots are available in Supplementary Fig. 10

represses the activation and nuclear translocation of the alternative NF-κB transcription factor, v-rel avian reticuloendotheliosis viral oncogene homologue B (RELB)[33–35].

RELB is emerging as an important player in various non-leukaemic cancers, including those dependent upon RAS signalling[26, 36, 37], yet has unclear functions or transcriptional targets in non-haematopoietic lineages[35, 38, 39]. Thus, we decided to prioritise investigation of the NDP52-TRAF3 interaction.

In HEK293T cells, FLAG-tagged TRAF3 co-immunoprecipitated with NDP52, via the cargo-binding zinc finger (ZnF) domain of the latter (Fig. 3b). Furthermore, an endogenous NDP52-TRAF3 complex could be immunoprecipitated from A549 cells (Fig. 3c). Thus, TRAF3 binds NDP52 in the mode of an autophagic cargo[13]. We next sought to determine via immunofluorescence whether the NDP52-TRAF3 complex was targeted to autophagic intermediates. As expected[40], most TRAF3 was found to target to the Golgi (Supplementary Fig. 3a). However, distinct TRAF3 foci localised with NDP52 and LC3B puncta, indicative of autophagosomal targeting (Fig. 3d). Furthermore, RNAi-mediated silencing of *NDP52* both abrogated the localisation of TRAF3 with LC3B foci (Fig. 3e) and increased endogenous TRAF3 protein levels (representative blot in Fig. 3f, quantified in Supplementary Fig. 3b). Pooled derivatives of A549 cells were generated where NDP52 was eliminated via CRISPR/Cas9. These also had elevated TRAF3 levels (Fig. 3g). Furthermore, TRAF3 targeting to lysosomes, the end-

point of autophagy pathway, was supported by the colocalisation of TRAF3 foci with the lysosomal marker LAMP2 after a brief treatment with the vacuolar H⁺ ATPase inhibitor, Bafilomycin A1 (BafA1) (Supplementary Fig. 3c).

The above data suggest that TRAF3 is degraded by autophagy. Readouts of low TRAF3 function include stabilisation of NF-κB inducing kinase (NIK), the subsequent processing of p100/NFκB2 to p52, which heterodimerises with RELB, and the eventual nuclear translocation of RELB (Supplementary Fig. 4). Supporting the hypothesis of TRAF3 degradation, TRAF3 protein levels were increased post-transcriptionally, and all of the above readouts were diminished, when autophagy was inhibited by RNAi against *ATG5* or *FIP200* in A549 cells (Fig. 4a, Supplementary Fig. 5a, b). The same differences were seen when comparing A549 Δ*ATG5* cells with control WT cells (Fig. 4b, c). Similar increases in TRAF3 levels and loss of p100 processing were also observed after *ULK1* RNAi (Supplementary Fig. 5c–e) or in *FIP200* deleted cells (Fig. 4d). *NDP52* or *FIP200* deletion also phenocopied *ATG5* loss in elevation of TGFβ-target gene transcript levels (Supplementary Fig. 5f) and inhibition of tumour growth in vivo (Fig. 4e). The *ATG5*-dependency of TRAF3 levels and transcriptional events were also demonstrated by CRISPR-Cas9 deletion of *ATG5* in a second RAS-mutant lung cancer line, NCI-H23 (Fig. 4f, Supplementary Fig. 6a). Furthermore, autophagy-dependent p100 processing, nuclear RELB, and the autophagic dependency of

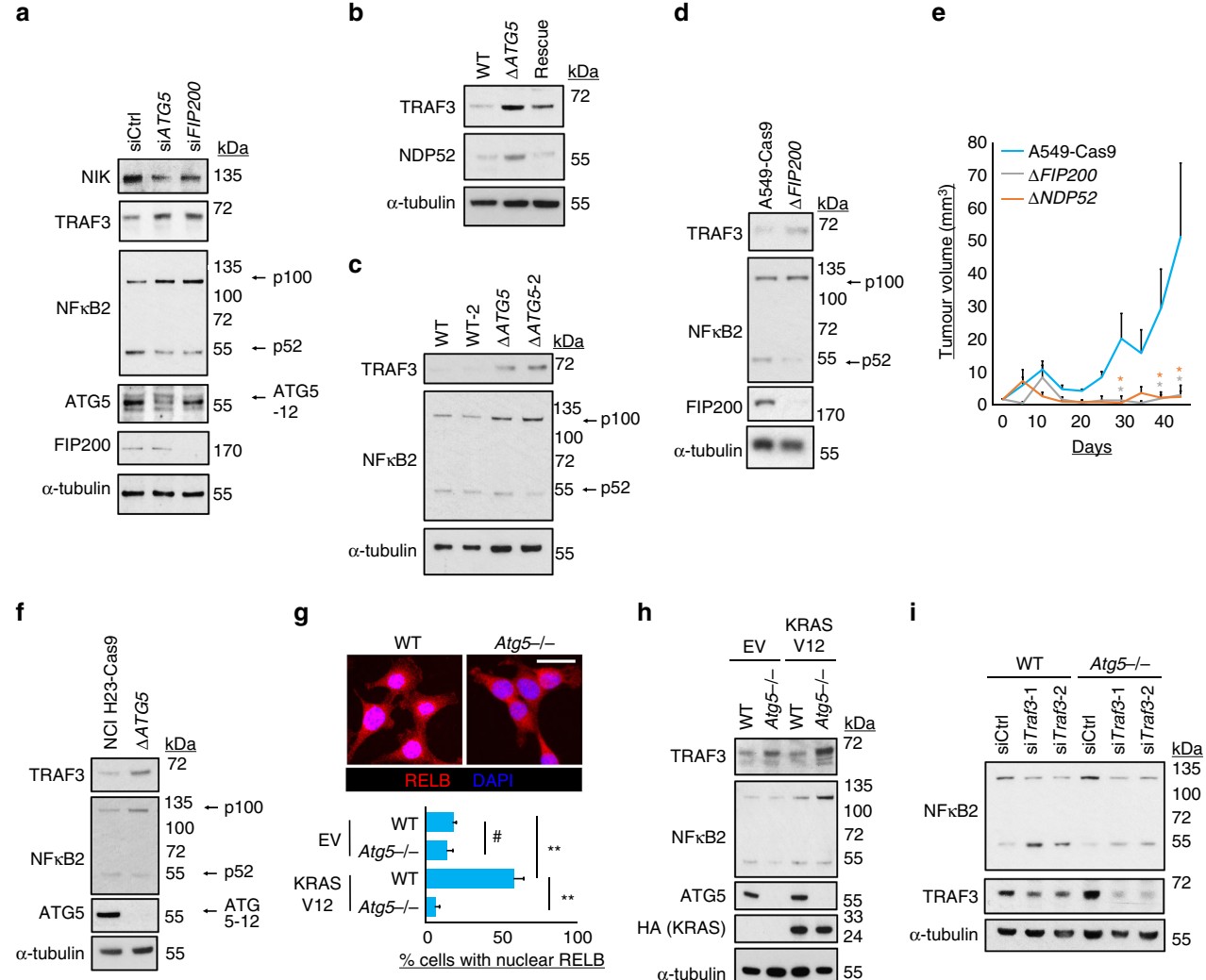

**Fig. 4** Alternative NF-κB signalling is stimulated by TRAF3 autophagy. **a** A549 cells were transfected with the indicated siRNA for 72 h and immunoblotted as shown (p100 and p52 indicate parental and processed NFκB2, respectively). **b**, **c** A549 WT, ΔATG5 and/or rescue cell clones were immunoblotted as shown. **d** A549-Cas9 control cells or pooled ΔFIP200 counterparts were immunoblotted as shown. **e** A549-Cas9 control cells or pooled ΔFIP200 and ΔNDP52 counterparts were subcutaneously injected into immunocompromised mice and tumour volume was measured longitudinally (means, $n = 8$ flanks, ±S.E.M., *$P < 0.05$ vs. A549-Cas9, two-tailed $t$-test). **f** NCI H23-Cas9 control cells or pooled ΔATG5 counterparts were immunoblotted as shown. **g**, **h** Wild-type (WT) or Atg5 null (Atg5−/−) empty vector (EV) or RAS-transformed (KRAS V12) MEFs were **g** stained for RELB or **h** immunoblotted. In **g**, wide-field images are shown (scale bar = 20 μm) and cells with nuclear RELB quantified (means, $n = 3$, ±S.E.M., **$P < 0.01$, #$P > 0.05$, ANOVA with Tukey's post hoc test). **i** WT or Atg5−/− MEF KRAS-V12 cells were transfected with the indicated siRNA for 32 h and then immunoblotted. Uncropped blots are available in Supplementary Fig. 10

TRAF3 regulation, were all apparent in MEFs, but only when these cells were transformed with RAS (Fig. 4g, h, Supplementary Fig. 6b, c). Thus, autophagy-mediated TRAF3 regulation and consequent RELB activity are strongly linked to RAS activation.

To show that autophagy-mediated turnover of TRAF3 is indeed the mechanism of regulation of NF-κB, we reverted TRAF3 levels using RNAi in Atg5−/− MEF KRAS V12 cells (Fig. 4i). This rescued p100 processing, indicating that blockade of NF-κB signalling due to loss of ATG5 was overcome. Knockdown of TRAF3 in A549 ΔATG5 cells also partially rescued growth in vivo (Supplementary Fig. 6d), although tumours did eventually regress. The eventual regression is in line with an additional observation made that complete loss of TRAF3 actually impairs tumorigenicity of A549 cells (Supplementary Fig. 6e). These data suggest that the elevated TRAF3 level in autophagy-deficient cells does suppress tumorigenicity. However, TRAF3 has other functions that contribute to long-term tumour growth. Notably, in non-RAS mutant murine cells there was

minimal evidence of TRAF3 turnover by potent autophagy stimuli such as amino-acid starvation (Supplementary Fig. 6f). Taken together, the above data show that autophagy is important in TRAF3 turnover downstream of RAS, which links to activation of nuclear RELB and tumorigenesis.

**RELB represses TGFβ-driven gene transcription**. Deletion of RELB phenocopies inhibition of autophagy in preventing growth of A549 cells in vivo (Fig. 5a, b). We speculated that RELB might be responsible for both the maintenance of tumorigenicity and the suppression of TGFβ downstream of autophagy. We further hypothesised that nuclear RELB might, directly or indirectly, repress transcription of TGFβ-driven genes. Thus, we performed gene expression profiling of A549 cells after silencing of RELB (Fig. 5c, Supplementary Data set 1). Strikingly, GSEA identified that, just as with inhibition of autophagy, the top-ranking correlation of the transcriptional signature of RELB deficiency was

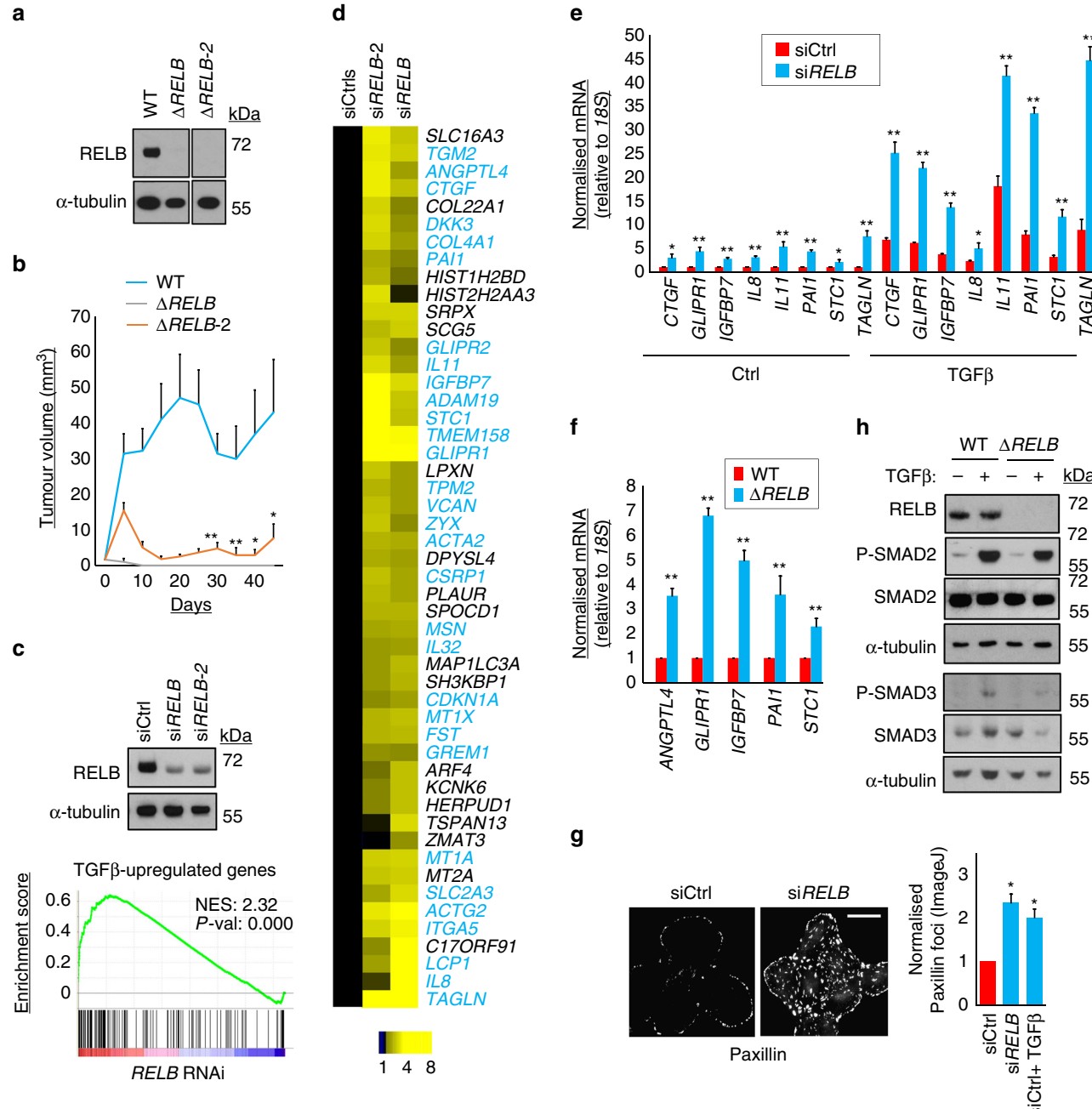

**Fig. 5** RELB promotes tumorigenesis and suppresses TGFβ target gene expression. **a** A549 ΔRELB cell clones (ΔRELB and ΔRELB-2) generated using CRISPR/Cas9 and two independent gRNA sequences were immunoblotted as shown. **b** A549 WT, ΔRELB and ΔRELB-2 cells were subcutaneously injected into immunocompromised mice and tumour volume was measured longitudinally (means, $n = 12$ flanks, ±S.E.M., *$P < 0.05$ or *$P < 0.01$ vs. WT, two-tailed $t$-test). **c** Two sequence-unrelated siRNA oligonucleotides targeting RELB were transfected in A549 cells (knockdown confirmed by immunoblotting). Expression profiling was performed vs. two sequence-unrelated non-targeting controls (Supplementary Data set 1). GSEA was performed, comparing genes differentially expressed in RELB knockdown cells with known oncogene and tumour suppressor profiles. The comparison yielding the greatest normalised enrichment score (NES), 'TGFβ upregulated genes', is shown here. **d** Fold-change heat map showing top 50 upregulated genes after RELB RNAi. Individual mean RELB siRNA values are normalised to the averaged means of the controls (siCtrls). Blue text indicates genes known to be transcriptionally upregulated downstream of TGFβ (Supplementary Table 2). **e**, **f** qRT-PCR was performed for the indicated transcripts after the following treatment regimens in A549 cells. **e** A549 cells were transfected with siCtrl or siRELB for 72 h. Cells were either left untreated (Ctrl) or treated with 5 ng/ml TGFβ1 for the final 16 h of transfection. **f** Exponentially dividing A549 WT and ΔRELB cells were compared. (means, $n = 3$, ±S.D., *$P < 0.05$ or **$P < 0.01$ vs. similarly treated siCtrl or WT cells, two-tailed $t$-test). **g** A549 cells were transfected with the indicated siRNA for 72 h, stained for paxillin and then imaged by widefield microscopy (scale bar = 20 μm). As a positive control, siCtrl cells were treated with 5 ng/ml TGF β1 for 8 h prior to fixation (+TGFβ). Paxillin foci were quantified on a per cell basis by image analysis (means, $n = 3$, ±S.E.M., *$P < 0.05$, two-tailed $t$-tests vs. siCtrl). **h** A549 WT or ΔRELB cells, with (+) or without (−) prior treatment with 5 ng/ml TGFβ1 for 16 h, were immunoblotted as shown (P-SMAD2 = phospho-S465/467 SMAD2, P-SMAD3 = phospho-S423/425 SMAD3). Uncropped blots are available in Supplementary Fig. 10

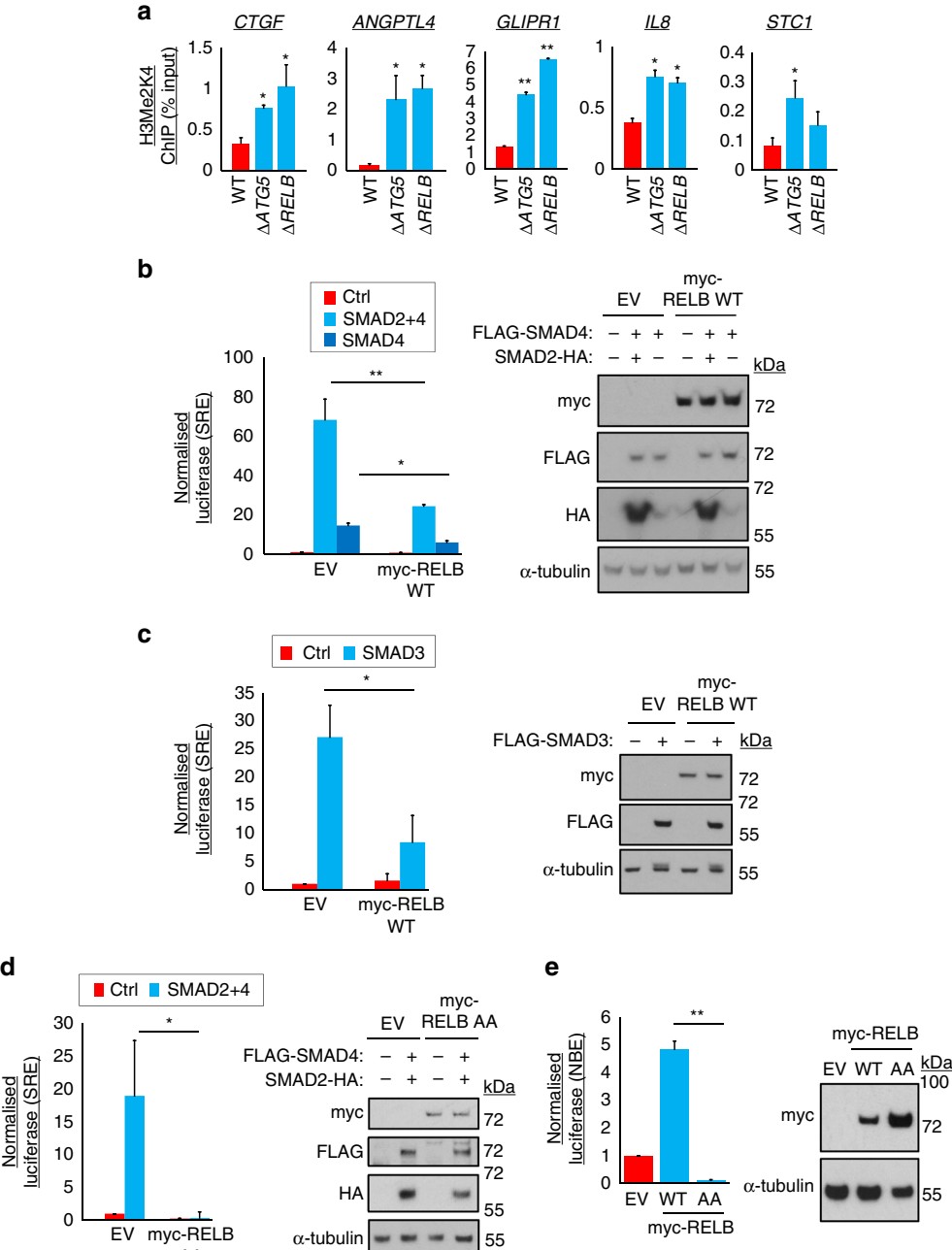

**Fig. 6** Autophagy/RELB suppress activation of TGFβ gene promoters independent of NF-κB binding elements. **a** WT, ΔATG5 and ΔRELB A549 cells were subjected to chromatin immunoprecipitation (ChIP) for dimethyl-K4-histone H3 (H3Me2K4). Precipitating proximal promoter DNA was quantified by qRT-PCR, expressed as a percentage of input (means, $n = 3$, ±S.D., *$P < 0.05$ or **$P < 0.01$ vs. WT cells, two-tailed $t$-test). Gene names are shown above the charts. **b–e** HEK239T cells were co-transfected for 30 h with a firefly luciferase reporter driven by **b–d** a SMAD-response element (SRE) or **e** an NF-κB binding consensus element (NBE), and a constitutive *Renilla* luciferase control plasmid, along with expression vectors for the indicated forms of epitope-tagged RELB (wild-type, WT, or R141A Y142A, AA), SMAD2, SMAD3 and/or SMAD4 or an empty vector (EV) control. Luciferase assays were then performed as described in Methods (*left*, firefly/*Renilla* ratios expressed in arbitrary units, means, $n = 3$, ±S.D., *$P < 0.05$ or *$P < 0.01$ vs. EV, two-tailed $t$-test). Immunoblotting was performed to assess transfected factor expression for each replicate (*right*, representative blots). Uncropped blots are available in Supplementary Fig. 10

with that of TGFβ-driven gene activity (Fig. 5c). Furthermore, concordance between the transcriptional changes seen upon autophagy inhibition and those evoked by RELB inhibition was evident upon heat map comparison of these expression profiles (Supplementary Fig. 7). Also consistent with the regulation of TGFβ output by RELB, 31 of the top 50 upregulated transcripts upon RELB inhibition were categorised as genes that are known to be upregulated upon TGFβ treatment (Fig. 5d, please consult

Supplementary Table 2 for detailed categorisation). qRT-PCR experiments directly confirmed the repression of many of these transcripts by RELB (Fig. 5e and Supplementary Fig. 8). TGFβ-driven genes thus identified were also upregulated upon ablation of *RELB* by CRISPR/Cas9 (Fig. 5f, ΔRELB cells). Finally, cytoskeletal responses characteristic of TGFβ activation were observed upon *RELB* silencing, namely the increased abundance of paxillin adhesion foci[41] (Fig. 5g).

The above data show that RELB is a repressor of TGFβ-driven gene activity. Importantly, similar to autophagy inhibition, RELB loss does not modulate the cytosolic TGFβ signalling pathway (Fig. 5h).

**RELB represses the activation of TGFβ target genes.** We next sought to identify a molecular mechanism by which RELB could reduce the abundance of TGFβ-driven transcripts. Firstly, we noted that the dimethyl-K4-histone H3 (H3Me2K4) chromatin

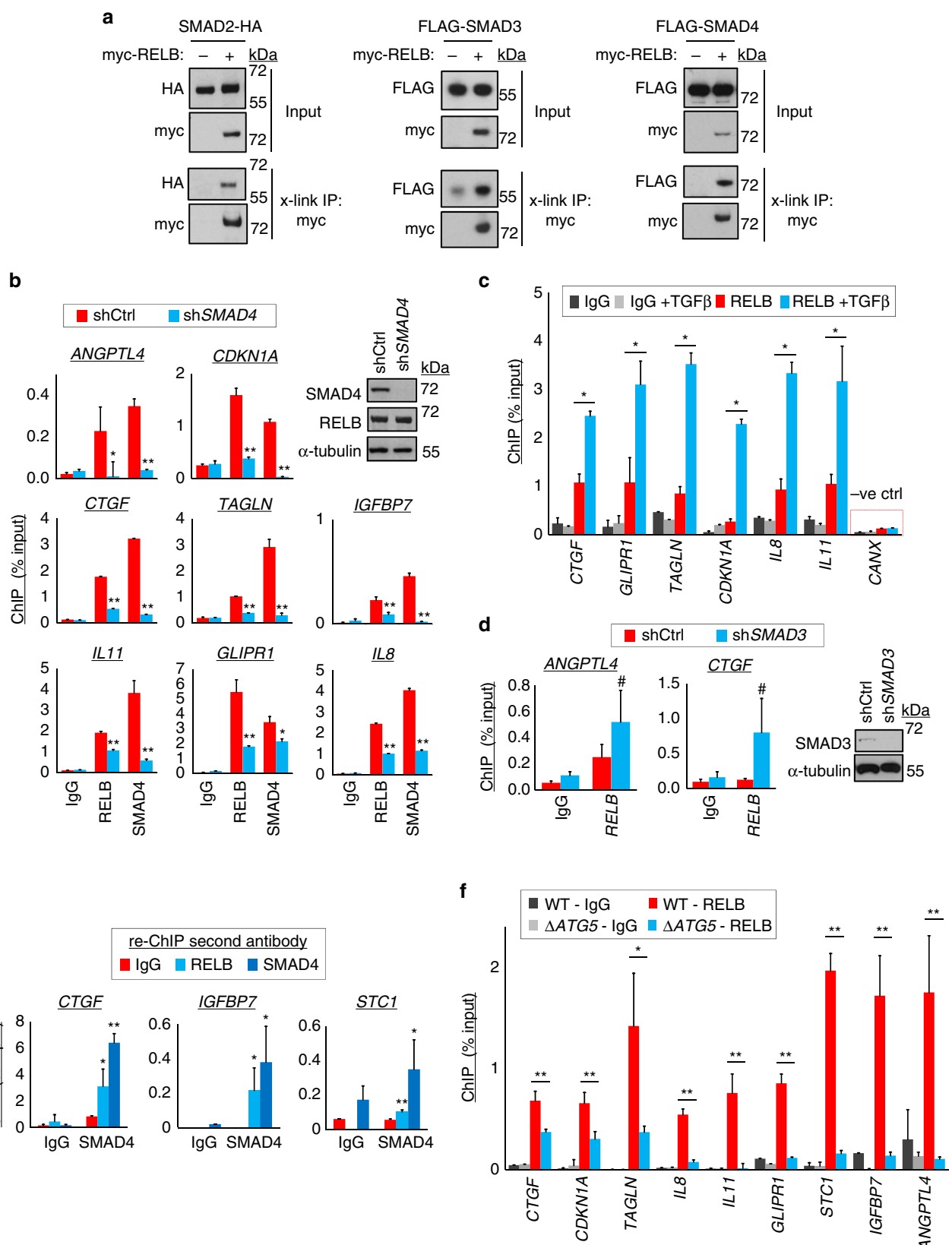

mark was upregulated at promoters for these genes in both $\Delta ATG5$ and $\Delta RELB$ cells (Fig. 6a), suggesting increased transcriptionally active or poised chromatin[42].

SMAD3-binding at SREs occurs via protein–DNA interaction[27]. Similarly, REL subunits of NF-κB classically bind to NF-κB-binding consensus elements (NBEs)[43, 44]. RELB is an outlier in the REL family, with a notably small set of bona fide target genes identified to date, particular in epithelial lineages[45]. Also, RELB can act as a transcriptional repressor[45–47]. Only a small subset of the RELB-regulated genes we identified, such as IL8 and IL11, contained validated NBEs for REL-family recruitment[48, 49]. Thus, we first tested the possibility that RELB might instead act on SREs to repress transcription, using luciferase reporter assays. Indeed, SMAD2- and SMAD4-driven activation of minimal SRE elements (bound directly by endogenous SMAD3) was diminished by transfection of wild-type RELB (Fig. 6b), as was direct SMAD3-driven transactivation (Fig. 6c). Strikingly, a RELB mutant deficient in DNA binding[50] (AA, R141A Y141A) was also effective in suppressing SRE activity (Fig. 6d) although, as expected, could not activate an NBE-driven reporter (Fig. 6e).

Taking the above data together, we conclude that RELB acts to inhibit promoter activity where SREs are present and undergoes an unconventional mode of recruitment that is independent of DNA-binding and NBE recognition.

**RELB interacts with SMADs in order to repress transcription.** We next hypothesised that RELB might form a protein–protein complex with SMAD(s). Confirming the potential for this, HEK293 cross-linking co-immunoprecipitation experiments detected the formation of RELB complexes containing SMAD2, SMAD3 or SMAD4 (Fig. 7a). We also assayed recruitment of endogenous RELB and SMAD4 to known SMAD-binding promoters using chromatin immunoprecipitation (ChIP). Indeed, we observed RELB occupancy of test promoters (Fig. 7b). SMAD4 also bound at these sites, as expected (Fig. 7b). Importantly, however, knockdown of SMAD4 reduced the promoter occupancy of both SMAD4 and RELB (Fig. 7b). This implied that recruitment of RELB was mediated via the protein–protein interaction with SMAD(s). Strengthening this, similar reductions in RELB binding were observed when promoter occupancy by SMAD4 was blocked with ALKi (Supplementary Fig. 9a). Conversely, RELB promoter occupancy was stimulated by exogenous TGFβ (Fig. 7c), which increases SMAD-2 and -4 binding (Supplementary Fig. 9b). As with SMAD4, knockdown of SMAD2 inhibited RELB binding to chromatin (Supplementary Fig. 9c). However, SMAD3 knockdown did not diminish RELB binding to

promoters, in fact producing an apparent increase in recruitment, albeit statistically insignificant (Fig. 7d). This suggests that SMAD2 and SMAD4 may be the key molecules for recruitment of RELB to endogenous promoters. Finally, re-ChIP (sequential ChIP) experiments showed that endogenous RELB and SMAD4 protein molecules co-occupied promoter DNA, consistent with recruitment of RELB via SMAD interaction (Fig. 7e). Last, we observed that indirect loss of RELB function in $\Delta ATG5$ cells also abrogated RELB recruitment to promoters (Fig. 7f).

Taking these data together, we conclude that SMAD protein complexes act as chromatin recruitment factors for RELB. Thus, RELB facilitates negative feedback on TGFβ-driven gene expression. Antagonism of SMAD-driven transcription occurs maximally when autophagy is active and acting to stimulate RELB function.

**SMAD4 inhibition rescues effects of autophagy loss in vivo.** TGFβ-driven transcription has anti-tumorigenic[27, 29] and pro-tumorigenic functions, depending upon context. Herein, we have discovered that autophagy antagonises SMAD-mediated transcription. In order to examine this in the context of a physiological readout, we employed the A549 $\Delta ATG5$ model. We stably suppressed SMAD4 expression using a short-hairpin RNA (shSMAD4) in both wild-type and $\Delta ATG5$ cells (Fig. 8a). This had no impact on autophagy, as assessed by LC3B lipidation status (Fig. 8a), or on cell proliferation in vitro (Fig. 8b). However, tumour growth kinetics were restored in vivo (Fig. 8c). Similar SMAD4-dependent suppression of tumour growth was seen upon FIP200 deletion in A549 cells (Fig. 8d, e). Thus, the suppression of TGFβ transcriptional output by autophagy and RELB/NF-κB is a component of the pro-tumorigenic effect of autophagy, at least in A549 cells. This provides a proof-of-principle that the capacity for regulation of transcriptional output by autophagy, seen in several cell lines, can affect cell behaviour in vivo. However, given the pleotropic effects of TGFβ, different aspects of tumour biology might be predominantly affected in other settings or models.

**Discussion**
We aimed to uncover new selective autophagy-based mechanisms acting to modify cytosolic signalling and cell fate. We focused on a model for RAS mutant cancer. In many such cancers, autophagy promotes tumorigenesis by mechanisms that are, at best, partially understood. Our findings are summarised in Fig. 9. A key component of the mechanism is that autophagy specifies transcriptional output. Previously, it has been shown that

**Fig. 7** RELB binds SMAD proteins and is recruited by SMAD(s) to promoters. **a** HEK293T cells were co-transfected with plasmids expressing myc-RELB and epitope-tagged SMAD proteins for 36 h. Cells were crosslinked and immunoprecipitated (x-link IP) using anti-myc antibodies. **b** A549 cells were stably transduced with control non-targeting shRNA (shCtrl) or shRNA targeting SMAD4 (shSMAD4). Immunoblotting was performed to confirm selective SMAD4 knockdown (upper right panel) and cells were subjected to ChIP analysis with RELB and SMAD4 antibodies (IgG, rabbit IgG negative control antibody). Promoter identities are above charts. Precipitating DNA abundance is expressed as a percentage of input (means, $n = 3$, $\pm$S.D., *$P < 0.05$ or **$P < 0.01$ shSMAD4 vs. cognate shCtrl IP, two-tailed t-test). **c** ChIP was performed using A549 cells, with or without prior treatment for 16 h with 5 ng/ml TGFβ1, using the antibodies indicated (IgG control or RELB). Promoter sequences analysed are given below (means, $n = 3$, $\pm$S.D., **$P < 0.01$, two-tailed t-test, negative ctrl = non-SMAD binding CANX promoter). **d** A549 cells were stably transduced with control non-targeting shRNA (shCtrl) or shRNA targeting SMAD3 (shSMAD3). Immunoblotting was performed to confirm selective SMAD3 knockdown and cells were subjected to ChIP analysis with RELB antibody (IgG, rabbit IgG negative control). Promoter identities are above the charts. Precipitating DNA abundance is expressed as a percentage of input (means, $n = 3$, $\pm$S.E.M., #$P > 0.05$ vs. cognate shCtrl IP, two-tailed t-test). **e** A549 cells analysed by re-ChIP to detect mutual binding of RELB and SMAD4 to promoters. The first round (initial ChIP) antibody (IgG or SMAD4) is indicated below the plots and the second round (re-ChIP) antibody (IgG, RELB or SMAD4) is shown in the colour key. Promoter identities are above individual plots (means, $n = 3$, $\pm$S.D., *$P < 0.05$ or **$P < 0.01$ vs. cognate IgG in second round of ChIP, two-tailed t-test). **f** A549 WT and $\Delta ATG5$ cells were subjected to ChIP with IgG or RELB antibodies (means, $n = 3$, $\pm$S.D., *$P < 0.05$ or **$P < 0.01$, two-tailed t-test). Uncropped blots are available in Supplementary Fig. 10

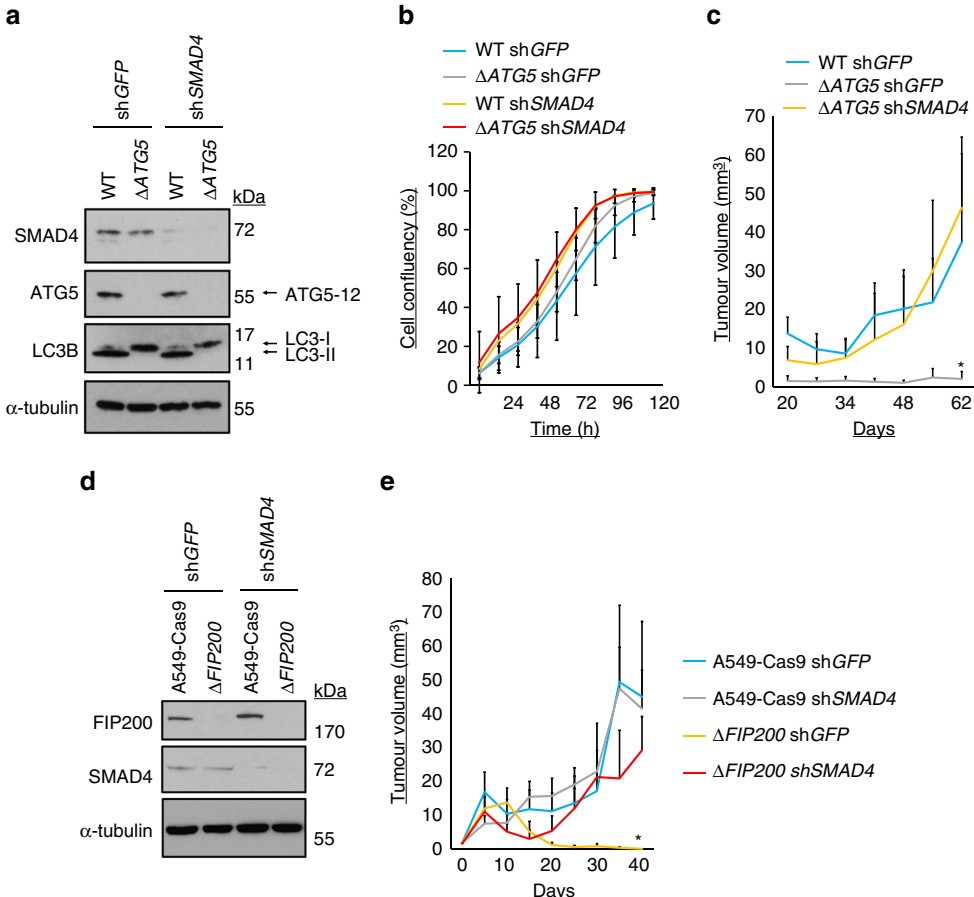

**Fig. 8** Inhibition of SMAD4 rescues the effects of autophagy loss in vivo. **a** A549 WT and ΔATG5 cells were transduced with control non-targeting shRNA (shGFP) or shRNA targeting SMAD4 (shSMAD4). Stable lines were immunoblotted as indicated. **b** Cells were plated for time-lapse phase-contrast videomicroscopy using an Incucyte microscope and cell proliferation was monitored by automated confluency analysis at set intervals post plating (n = 9 wells, ± S.D.). **c** Cells were subcutaneously injected into immunocompromised mice and tumour volume was monitored longitudinally (means, n = 10 flanks, ±S.E.M., *P < 0.05 vs. WT shGFP, two-tailed t-test). **d, e** A549-Cas9 and cognate A549 ΔFIP200 cell lines were transduced with shGFP or shSMAD4. Selected pools were **d** immunoblotted as indicated or **e** subcutaneously injected into immunocompromised mice, whereupon tumour volume was monitored longitudinally (means, n = 8 flanks, ±S.E.M., *P < 0.05 vs. all other conditions, two-tailed t-test). Uncropped blots are available in Supplementary Fig. 10

transcriptional responses to interferon pathway stimuli are exaggerated when autophagy is inhibited[9]. However, the most penetrant effect we observed upon autophagy inhibition was upregulation of the transcriptional response to TGFβ signalling.

TGFβ-driven transcription acts to suppress tumour establishment and growth in a number of cancer types, via a range of modalities encompassing senescence, inhibition of proliferation and engagement of cell death[27, 29]. Indeed, components of the TGFβ pathway such as SMAD4 and the ligand receptor TGFBR2 are encoded by classic tumour suppressor genes, which undergo direct loss-of-function mutation during tumorigenesis in certain cancers[27]. We reveal an alternate mechanism for suppression of TGFβ function. RAS engages selective autophagy, which in turn mediates signalling events resulting in antagonistic cross-talk with TGFβ, dampening its tumour-suppressive transcriptional output and thus permitting tumorigenesis in vivo. The molecular basis via which upregulated TGFβ signalling inhibits tumour growth remains to be explored. It is likely well-understood mechanisms such as inhibition of cell-cycle progression and apoptosis play a role. In RAS-driven cancers, cell-autonomous autophagy function can promote lipid catabolism and regulate the balance between oxidative phosphorylation and aerobic usage of glucose[17, 18, 20, 24]. It is possible that regulation of TGFβ output could affect metabolism, although this remains to be investigated.

In certain contexts, TGFβ/SMAD can promote EMT and increased tumour aggression[27, 51]. Some EMT-implicated genes are upregulated when autophagy is inhibited in the system described herein, including CDH2 (N-Cadherin) and SNAI2 (SLUG) (Supplementary Fig. 7). This is also thematically consistent with reports that autophagy can post-transcriptionally repress the levels of transcription factors that would otherwise promote EMT, such as TWIST and SNAIL[52, 53]. Although, on the other hand, autophagy has been shown in HRAS-transformed mammary epithelial cells to promote the secretion of factors that facilitate matrix invasion[54]. Thus, it is possible that other in vivo model systems will reveal a more complex relationship between autophagy and the outcomes of TGFβ signalling, including regulation of pro-tumorigenic EMT events.

The mechanism of selective autophagy involvement in regulation of gene expression was revealed as termination of the activity of a new cargo, TRAF3. This culminates in nuclear translocation of the transcription factor RELB. There is a growing consensus that such 'signalphagy'-type responses will prove important in physiological autophagy function. Indeed, TRAF3 degradation joins recent examples such as degradation of GSK3β[5] and β-catenin[8], TBK1[9] and IRF3[10], by autophagy in various systems.

We uncovered a new function for RELB in repression of TGFβ transcriptional output, wherein RELB operates directly upon

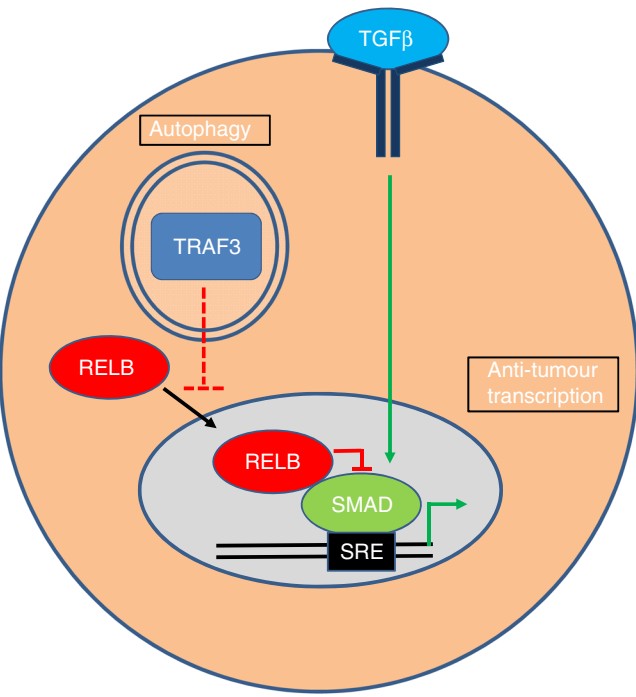

**Fig. 9** Model of the molecular events comprising an autophagy-mediated, inhibitory cross-talk with the TGFβ pathway in RAS-transformed cells. Selective autophagy of TRAF3 via cargo receptors, such as NDP52, terminates the tonic inhibition of the alternative NF-κB pathway that is ordinarily observed in unstimulated primary cells. This results in nuclear activity of RELB. Downstream of TGFβ, frequently present in the tumour milieu, DNA binding by SMADs drives gene transcription. Directly TGFβ-responsive gene promoters recruit SMAD complexes to SMAD-response elements (SREs). However, RELB has the ability to repress SRE-containing gene promoters. This occurs not via interaction with NF-κB consensus sites but instead via recruitment to chromatin by protein–protein interaction with active SMAD complexes. Thus, transcriptionally repressive RELB effectively 'hijacks' SMAD promoters to exert negative feedback on TGFβ-mediated transcription. In the absence of autophagy, the above events involving RELB are not engaged. Thus, in vivo, autocrine and/or paracrine sources of TGFβ repress tumorigenesis when autophagy is ablated. Please note though, that in in vivo models other than A549 cells, it is possible that altered sensitivity to TGFβ could potentially have other phenotypic outcomes

SMAD-binding promoters. Thus, TGFβ- and RAS-mediated signalling, the latter involving autophagy, converge antagonistically within the nucleus. Upon delving further into the mechanism by which RELB could regulate transcription, we discovered that it was not acting in a conventional manner on NF-κB-binding consensus sequences. We instead elucidated a new action for RELB in binding to SMAD proteins in the nucleus and thus 'hijacking' TGFβ-activated promoters. We note that a recent set of reports has suggested that TGFβ can activate LC3 expression and lipidation, and thus autophagic flux, in some cell types[55, 56]. This is theoretically compatible with our findings—autophagy stimulation would fit into our model as a negative feedback loop in TGFβ signalling. However, we did not find that SMAD knockdown affected LC3B levels or lipidation.

In summary, this work reveals a new contribution of autophagy to tumorigenesis in vivo via a form of selective autophagy, or signalphagy, which results in reprogramming of gene expression. Discoveries of the signal regulatory functions of autophagy, such as this, give important insight into the range and mechanistic diversity of selective autophagy pathways in regulating cell

physiology. Increased comprehension of signal regulatory functions of autophagy will lead to future improvements in understanding and targeting autophagy outcomes in health and disease.

## Methods

**Cells and materials**. ALK2/4/5 inhibitor (2-(3-(6-methylpyridin-2-yl)-1H-pyr-azol-4-yl)-1,5-naphthyridine) was from Calbiochem (#616452). Recombinant human TGFβ1 ligand was from AbD Serotec. All cell lines were cultured in standard DMEM supplemented with 10% foetal calf serum and penicillin/strep-tomycin, at 37 °C and 5% $CO_2$. A549-EcoR cells are A549 cells expressing the Ecotropic receptor for retroviral infection and G418 resistance marker. They were obtained from the laboratory of Chris Marshall (Institute of Cancer Research, London). A549-EcoR cells were identity checked by microsatellite genotyping. NCI-H23 parental cells were obtained from ATCC and derivatised to express Ecotropic receptor. Neither A549 nor NCI-H23 cell lines are commonly mis-identified (source: ICLAC database v8, updated December 2016). Phoenix-Eco cells were provided by Kevin Ryan, Beatson Institute, UK. HEK293T cells and deriva-tives are commonly misidentified due to errors during dissemination from one laboratory to another. However, our cells are the established HEK293FT substrain, originally obtained from a commercial vendor (Clontech). MEFs were provided by Noboru Mizushima[57]. MEFs were immortalised via retroviral transduction of a derivative of pSIRIP shRNA p19-2—originally a gift from Tyler Jacks, Addgene #14090[58]—wherein the selection marker was altered to confer resistance to hygromycin. Immortalised MEFs were transformed with retrovirus produced from the MSCV NTAP $KRAS^{G12V}$ plasmid. A549 FLAG-HA-TRAF3 lines were derived by infection with retrovirus produced from cognate MSCV NTAP plasmids and selection in puromycin. The NTAP tag is a tandem affinity FLAG-HA tag. pBabe-puro GFP-ATG5-derived retrovirus was used to rescue A549 ΔATG5 cells. To obtain SMAD4 knockdown cells, pLKO.1 shNTC, shGFP and/or shSMAD4[59] plasmids were packaged in HEK293T cells by co-transfection with pMD2.G and psPAX2 plasmids, and used to stably infect A549 cell lines. pLKO.1 viruses were puromycin resistant except for either TRAF3 knockdowns or any infection of A549-Cas9 and derivates, where, in both instances, pLKO.1-hygro viruses were used. A549-Cas9 cells were generated by infection of A549 cells with lentivirus generated from lentiCas9-BLAST and selection for blasticidin resistance. To obtain SMAD3 knockdown cells, retrovirus produced from pRetroSuper shCtrl or shSMAD3 plasmids were used to infect A549 cells and stable pools of cells selected in puromycin. All cell lines used during these studies tested negative in bi-monthly mycoplasma screening.

**Antibodies**. All antibodies are fully described in Supplementary Table 3 and were routinely used at 1:2000 dilution for immunoblotting and 1:200 dilution unless noted otherwise. A volume of 5 µl of antibody was routinely used per condition in protein and chromatin immunoprecipitation experiments.

**Plasmids**. Plasmids used in this study are fully described in Supplementary Table 4.

**Mass spectrometry**. Four 15 cm cell culture dishes of A549-EcoR cells stably expressing NTAP-NDP52 were washed and harvested with ice-cold PBS followed by storage at −80 °C or immediate lysis in 4 ml MCLB buffer. Cell debris was removed from the lysates by centrifugation and supernatants were passed through 0.45 µm spin filters (Millipore). Anti-HA-agarose (60 µl slurry, Sigma) was added to lysates for immunoprecipitation overnight at 4 °C, rotating. Samples were washed five times with 1 ml MCLB followed by five washes with PBS and elution with 150 µl HA peptide (250 µg/ml, Sigma). Eluted immune complexes were essentially processed in a similar manner to those in published studies[60, 61]. Briefly, proteins were precipitated with trichloroacetic acid (Sigma) followed by digestion with trypsin (Promega) and desalting by stage tips. Samples were analysed in technical duplicates on a LTQ Velos (Thermo Scientific). Spectra were identified by Sequest searches followed by target-decoy filtering and linear discriminant analy-sis[62]. Peptides that could be assigned to more than one protein in the database were assembled into proteins according to parsimony principles. For CompPASS ana-lysis, we employed 34 unrelated bait proteins that were all previously processed in the same way in A549 cells. Weighted and normalised D-scores (WDN-score) were calculated based on average peptide spectral matches (APSMs). Proteins with WDN ≥ 1 and APSM ≥ 3 were considered as high-confident candidate interacting proteins (HCIPs). Proteins with APSM ≥ 2 and that had interactions documented in BIOGIRD with at least two of the HCIPs and/or NDP52 were also considered candidate interactors (subthreshold).

**Luciferase assays**. HEK293T cells were transfected in triplicate with Lipofecta-mine 2000 according to the manufacturer's instructions. After 36 h, cells were lysed in Passive Lysis Buffer (Promega) for 15 min at room temperature and both firefly and Renilla luciferase activity measured using the Dual-Luciferase® Reporter Assay System (Promega) and a Fluoroskan Ascent FL plate reader (Labsystems), fol-lowing manufacturer's instructions. Cells were transfected with plasmids described in Figure legends as well as an identical replicate set in which pGL3 basic was

substituted for the reporter construct to confirm negligible firefly luciferase activity in the absence of specific binding elements from the reporters.

**Immunoblotting**. Cells were routinely lysed in RIPA lysis buffer (50 mM Tris-HCl, pH 7.5, 0.5% Na deoxycholate, 1% Triton-X-100, 150 mM NaCl, complete protease inhibitors + 1 mM EDTA (Roche #05056489001), 2 mM Na orthovanadate, 25 mM Na β-glycerophosphate, 10 mM NaF, 10 mM Na pyrophosphate). Cells were washed briefly in ice-cold PBS and pre-incubated with ice-cold RIPA lysis buffer for 5 min prior to scraping and homogenisation. Scraped and homogenised lysates were incubated on ice for a further 10 min before clarification by centrifugation in a benchtop centrifuge at full-speed for 15 min at 4 °C. The supernatant was removed and quantified by Pierce BCA Protein Assay (Life Technologies #23225). For immunoblotting to detect NIK, RIPA lysis buffer was supplemented with 20 μM MG132 to stabilise NIK levels during lysis. Gel electrophoresis was performed using 4–12% NuPAGE Novex Bis-Tris gels, or for LC3B immunoblotting, 4–20% NuPAGE Tris-Glycine gels. Nitrocellulose membranes, or PVDF membranes for LC3B immunoblotting, were probed according to the standard methods and visualised using enhanced chemiluminescence.

**Immunoprecipitations**. For tagged protein immunoprecipitation, HEK293T cells were transfected with Lipofectamine 2000 according to the manufacturer's instructions. After 24 h, cells were lysed in IGEPAL buffer (50 mM Tris-HCl, pH 7.5, 0.5% IGEPAL CA-630, 150 mM NaCl, complete protease inhibitors + 1 mM EDTA (Roche), 2 mM activated orthovanadate, 25 mM Na β-glycerophosphate, 10 mM NaF, 10 mM Na pyrophosphate). For cross-linking immunoprecipitation experiments, 24 h after transfection, cells were cross-linked with 1% para-formaldehyde in DMEM for 10 min at room temperature, and quenched by 0.125 M glycine for 10 min. Cells were washed twice with ice-cold PBS, scraped, pelleted and lysed in 1% SDS, 10 mM EDTA and 50 mM Tris, pH 8.0 + protease inhibitors, then sonicated. Samples were then clarified by centrifugation and supernatants diluted with IGEPAL buffer to 0.1% SDS. For both standard and cross-linking immunoprecipitations, lysates were incubated with rabbit anti-myc-conjugated agarose beads (Sigma) for 4 h and then beads were washed three times in IGEPAL and eluted in Laemmli sample buffer.

For endogenous co-immunoprecipitation, $4 \times 10^6$ A549 cells were seeded overnight in 15 cm dishes and lysed in IGEPAL buffer. Magnetic beads were conjugated with antibody using the Dynabead coupling kit (Invitrogen #14311d), following manufacturer's instructions. Lysates were incubated with these beads for 4 h, washed three times in IGEPAL buffer and eluted in 4% SDS (no reducing agent).

**Light microscopy**. For immunofluorescence, cells were grown on glass coverslips and fixed in 4% paraformaldehyde for 10 min at room temperature and permeabilised with 0.25% Triton X-100 for 20 min at room temperature, or, for LC3B immunofluorescence, with methanol for 5 min at −20 °C. Cells were incubated with primary antibodies overnight at 4 °C and secondary antibodies and DAPI for 1 h at room temperature and mounted using DAKO fluorescent mounting medium. Secondary antibodies used were Invitrogen goat anti-mouse, anti-rat and anti-rabbit antibodies conjugated to Alexa Fluor 488, 594 or 647. Widefield fluorescence images were captured with an Olympus BX51 microscope and an Olympus DP71 camera using cell^F software (Olympus Soft Imaging Solutions GmbH v2.8). Acquisition time and illumination intensity was consistent across experimental conditions. RELB nuclear localisation was quantified by single blind scoring. Confocal microscopy was performed using an Olympus FV1000 confocal microscope using Olympus proprietary software (Fluoview). Images were viewed in Image J using the Bio-Formats v5.1 plugin. Acquisition parameters were consistent across experimental conditions. Brightness and contrast were adjusted consistently across experimental conditions using Image J software. For merge images for colocalisation, Paint.NET software (v 3.3x, 3.5) was used to adjust the relative level and curves for red, green or blue channels and changes applied to the whole image, the same changes were applied to all images across an individual experiment. Colocalisation of punctate signals in different channels was quantified by single blind scoring. For quantification of Paxillin foci, the Foci Picker 3D plugin for Image J was used. The number of foci was normalised to cell number and the criteria for scoring was unchanged across experimental conditions. In all experiments where quantification was performed, a minimum of 100 cells were scored for each condition and biological replicate.

**Immunohistochemistry**. For immunohistochemistry, xenograft tumours were dissected from the mouse immediately after killing and fixed in 10% neutral-buffered formalin overnight. Tumours were transferred into 70% EtOH before embedding in paraffin and sectioning. All further processing was performed using a BOND III immunostainer (Leica Biosystems). Dewaxing was performed using dewaxing solution (Leica #AR9222), epitope retrieval was performed using solution 1 (Leica #AR99961). Samples were subjected to DAB immunohistochemistry using the BOND Refine Kit (Leica #AR922) with the exception of blocking, which was performed using the mouse IgG blocking solution (Vector #MKB2213).

**siRNA transfection**. $10^5$ A549 or $7.5 \times 10^4$ MEFs was seeded overnight in 35 mm diameter wells. Cells were transfected for 8 h with Oligofectamine (Life Technologies #12252-011) and 50 pmol of siRNA, according to the manufacturer's instructions. Identities along with bracketed sequences/catalogue numbers of the siRNAs are as follows: Ctrl(sequence not provided by manufacturer, Dharmacon # D-001210-01), Ctrl-2 (AATTCTCCGAACGTGTCACGT, Qiagen #SI1027310), Traf3 (TGCAATCGTTGTTTCAAATATA, Qiagen #SI01454551), Traf3-2 (AAGGTTTCATTTGGTATTTAT, Qiagen #SI01454558), ATG5 (CATCT-GAGCTACCCGGATA, Dharmacon #004374-03), FIP200 (CTGGGACGGATA-CAAATCCAA, Qiagen #SI02664571), ULK1 (sequence not provided by manufacturer, Hs_ULK1_5 siRNA, Qiagen), NDP52 (AAGATGAAACTTA-CACTACTT,Qiagen #SI0431794), RELB (CACAGATGAATTGGAGATCAT, Qiagen #SI03038483), RELB-2 (CAGCTACGGCGTGGACAAGAA, Qiagen #SI05001451), NFKB2 (AACCCAGGTCTGGATGGTATT,Qiagen #SI00300965).

**Transcriptomic expression profiling**. RNA was harvested from cells 72 h after transfection with siRNA, and total RNA was purified using the RNeasy mini kit (Qiagen), including optional DNA digestion. Biotin-labelled cRNA was prepared using the TotalPrep RNA Amplification Kit (Ambion, #AMIL1791), according to the manufacturer's instructions, and hybridised to the Illumina HT-12 human bead array v4.0. The array was scanned using the Illumina HiScan platform. Raw data was processed using VST transformation and subsequent RSN normalisation, using the lumi package in R[63, 64].

All downstream analyses were also performed in R. Firstly, subthreshold probes were identified after reverse transformation of the normalised probe intensities, using the lumi package functions inverseVST and detectionCall (threshold = 0.01). These were excluded from subsequent analyses. To identify differing probe intensities upon autophagy inhibition, unpaired $t$-tests were performed between all six non-targeting control samples (three biological replicates each of siCtrl and siCtrl-2) and between all six autophagy gene siRNA samples (three biological replicates each of siATG5 and siULK1), using the RSN-normalised values. Correction for multiple comparison was made using p.adjust and the FDR (false discovery rate) method at the 0.1 level. Values for selected probes were then reverse transformed for presentation in Tables and heat maps herein. Where fold cut-offs were employed as a criterion for selection, these were based upon the fold difference in the mean of all six autophagy siRNA samples and all six siCtrl samples, in the reverse transformed data set. Regardless of whether testing significance or fold changes, all selected probes were also required to meet the criterion that the average intensities of both subsets of control siRNA in the reverse-transformed data set were unidirectionally altered when compared pairwise with each individual test siRNA set (for example, siCtrl < siATG5 and siCtrl2 < siATG5 and siCtrl < siULK1 and siCtrl-2 < siULK1). In Figs. 2b and 5d, probe intensities when more than one probe was selected for a given gene were collapsed by averaging the fold changes for each siRNA, such that each gene only appeared once in the final list of 50.

Identical methodology to that above was used to determine significant changes and/or fold-change parameters upon alternative NF-κB inhibition (three biological replicates each of siRELB and siRELB-2).

All heatmaps in the manuscript were produced using heatmap.2[65]. R scripts used for data processing are available upon request from the corresponding author.

**Gene set enrichment analysis**. We acknowledge our use of the gene set enrichment analysis, GSEA, software and the Molecular Signature Database (MSigDB) at http://www.broad.mit.edu/gsea/32. Gene set enrichment was performed using normalised, reverse transformed data sets with low-intensity probes filtered out, as described above. Six control replicates were compared against six ATG5/ULK1 siRNA replicates or six RELB replicates in each of two sets of analyses. The Gene Set used for comparison was 'oncogenic signatures' from MSigDB. The analysis was performed with the following parameters: probes collapsed to single-gene identities and 1000 permutations, permuting on gene set and Signal2Noise selected.

**qRT-PCR**. Total RNA was extracted from cells using the RNeasy mini kit (Qiagen) with QIAshredder columns, following manufacturer's instructions. cDNA was synthesised using 1–5 μg template RNA using the First Strand cDNA synthesis kit (Applied Biosystems). qRT-PCR was performed with DyNAmo HS SYBR Green qPCR mastermix (Thermo Scientific F-410) on a Rotor-Gene RG300 (Corbett Research) or a StepOne PlusReal-Time qPCR machines and analysed with the corresponding software. All experiments were quantified in relation to standard curves where presented with an individual gene per chart, and readings were normalised to 18S levels. In higher-throughput analyses, multiple genes are presented on the same chart and this indicates that ΔΔCt calculation methodology was used, employing 18S as reference. Primers are described in Supplementary Table 5.

**Chromatin immunoprecipitation**. Samples were cross-linked with 1% paraformaldehyde in DMEM for 10 min at room temperature and quenched by 0.125 M glycine for 10 min. Cells were washed twice with ice-cold PBS, scraped, pelleted

and lysed in 1% SDS, 10 mM EDTA and 50 mM Tris, pH 8.0 + protease inhibitor tablets (Roche), then supernatants were sonicated uniformly to generate fragments ranging from 200 to 1000 bp. A sample was stored as input. Sonicated supernatants were pre-cleared with Protein A Dynabeads (Life Technologies) bound with 5 µg rabbit IgG (Cell Signaling Technology) in the presence of 2 µg salmon sperm DNA. Pre-cleared lysates were incubated with Protein A Dynabeads (Invitrogen) and 5 µg of immunoprecipitating antibody or rabbit IgG control, overnight at 4 °C under rotation. Washes were performed in each of the buffers sequentially: low salt buffer (0.1% SDS, 1% Triton X-100, 2 mM EDTA, 20 mM Tris, 200 mM NaCl, pH 8.1), high salt buffer (0.1% SDS, 1% Triton X-100, 2 mM EDTA, 20 mM Tris, 550 mM NaCl, pH 8.1), LiCl buffer (250 mM LiCl, 1% IGEPAL, 1% sodium deoxycholate, 1 mM EDTA and 10 mM Tris, pH 8.1) and twice in Tris-EDTA (10 mM Tris-HCl 1 mM EDTA, pH 8.1). Beads were incubated and then vortexed, twice sequentially, in elution buffer (100 mM NaHCO$_3$ and 1% SDS). 200 mM NaCl was added to eluted, precipitated chromatin, or to input samples, and cross-linking was reversed by incubation at 65 °C overnight. The sample was then adjusted to 40 µg/ml Proteinase K, 40 mM Tris, 10 mM EDTA and digestion was performed for 4 h at 45 °C. DNA was purified with a Qiagen PCR clean-up kit according to the manufacturer's instructions, and samples were quantified for specific DNA species by SYBR-green qRT-PCR as described above, using the tabulated primers (Supplementary Table 5).

For re-ChIP assays, following the first immunoprecipitation (as above), beads were washed three times with re-ChIP wash buffer (2 mM EDTA, 200 mM NaCl, 0.1% SDS, 1% NP-40) and twice with Tris-EDTA buffer. DNA was eluted in Re-ChIP elution buffer (10 mM Tris HCl, 1 mM EDTA, 2% SDS) for 30 min at 37 °C. Following elution, the supernatant was diluted to a concentration of 0.1% SDS with ChIP dilution buffer (1% Triton-X-100, 2 mM EDTA, 150 mM NaCl, 20 mM Tris-HCl, pH 8.1) supplemented with 50 µg bovine serum albumin and protease inhibitor tablets (Roche). Samples were incubated with Protein A Dynabeads (Invitrogen) and 5 µg of immunoprecipitating antibody or rabbit IgG control, overnight at 4 °C under rotation. Beads were washed, DNA eluted and reverse crosslinking performed as in the standard ChIP assay. Primers used for qRT-PCR analysis are shown in Supplementary Table 5.

**CRISPR/Cas9-mediated gene editing in A549 cells.** For clonal cell cultures, cells were transfected with a 50:50 ratio of gRNA plasmid and Cas9:puro2A by nucleofection with Lonza Nucleofector kit T, according to the manufacturer's instructions for A549 cells. Cells were selected for successful transient transfection with 2.5 µg/ml puromycin, 24 h post transfection for a further 24 h duration, and then grown in regular medium as single cell colonies. Control colonies were derived after transfection of empty gRNA plasmid. ΔATG5 clones were derived after transfection of gRNA targeting GAGATATGGTTTGAATATGA. ΔRELB clones were derived from gRNAs targeting GCCACGCCTGGTGTCTCGCG (clone 1) or GATCATCGACGAGTACATCA (clone 2).

For pooled selections, A549-EcoR and NCI-H23-EcoR cell populations were derivatised by infection with lenti-Cas9 Blast lentivirus packaged in HEK293T cells (see Cells and materials)[26] and selection in 30 and 15 µg/ml blasticidin, respectively. These cells were further transduced with unmodified lentiGuide-puro lentivirus, to provide control pooled cells (referred to as A549-Cas9 and NCI-H23-Cas9 in data figures), or with lentiGuide-puro expressing specific gRNA sequences. For A549-Cas9 editing these sequences were CTGGTTAGGCACTCCAACAG (FIP200), TGAGTATTACACCTTCATGT (NDP52) or GAAAGACCTGCGAGACCACG (TRAF3). For NCI-H23-Cas9, the sequence was GAGATATGGTTTGAATATGA (ATG5). All cells were selected in 2 µg/ml puromycin.

**Animal models.** Female CD1-nude mice were subcutaneously injected on both flanks with 100 µl of $1.5 \times 10^7$ (Figs. 1e and 8c only) or $3.5 \times 10^7$ cells/ml A549 suspension in Hank's Buffered Saline Solution. Minimally, eight flanks were injected in order to achieve sufficient experimental power, based upon prior experience of the model. Mice were from Charles River Laboratories and in all experiment groups were identically between 2 and 5 months of age and mice allocated to groups from purchased stocks at random. Mice were housed in individually ventilated cages and tumour measurements thereafter taken using callipers. Investigators were blinded to mouse identity to the extent that mice were ear-tagged with code numbers to identity to which experimental groups they belonged. Nonetheless, mice in individual experimental groups were housed together in the same cage. Mice were killed if tumour length exceeded 10 mm or ulceration was observed. At endpoint, tumour tissue was fixed overnight in 10% neutral-buffered formalin and paraffin embedded. All animal studies were performed after University of Edinburgh local ethical review and under the authority of a UK Home Office project licence.

**Statistics.** All replicates are biological replicates unless explicitly indicated otherwise in Figure legends. All statistical tests were based upon estimates of variation relating to presumed normal distribution (standard deviation and/or standard error of the mean). Unless stated otherwise in the Figure legend, for example where multiple testing was corrected for, statistical analyses were standard or one-sample Student's t-tests. All t-tests were two-tailed. As a special instance, treatment of gene expression profiling is described in a dedicated Method entry, above.

Blinding was performed where practicable and in such instances is described under individual methodology descriptions (for example, immunofluorescence scoring).

**Code availability**. R scripts used to process array data are available from the authors.

**Data availability**. A MIAME-compliant data set encompassing all the array-based expression profiling raw data, processed data, and further experimental detail, has been deposited in NCBI Gene Expression Omnibus (http://www.ncbi.nlm.nih.gov/geo/), with accession number GSE73158. Raw data for NDP52 affinity purification mass spectrometry have been deposited at MassIVE (http://massive.ucsd.edu) with accession number MSV000081221. Other data are available from the authors upon reasonable request.

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

## Acknowledgements

We thank Andrew Finch, Noboru Mizushima and Ian Ganley for generous gifts of MEFs. We also thank Morwenna Muir and Val Brunton for their assistance with xenograft experiments. A.C.N. was funded by a Medical Research Council (UK) graduate studentship. This work was supported by Cancer Research UK in the form of a Career Development Fellowship to S.W. (C20685/A12825) and C.B. was supported by the Deutsche Forschungsgemeinschaft (German Research Foundation) within the framework of the Munich Cluster for Systems Neurology (EXC 1010 SyNergy).

## Author contributions

A.C.N., A.J.K. and S.W. designed and interpreted the expression array analyses. C.B. and S.W. designed and analysed the mass spectrometry screen. C.B. performed the mass spectrometry. A.C.N. and S.W. designed and interpreted all other experiments. A.C.N. and S.W. performed xenograft analyses. Y.D. performed the ChIP analyses.

## Additional information

**Competing interests:** The authors declare no competing financial interests.

