## [Peer Review File · Nature Communications]

Reviewers' comments:

Reviewer #1 (Remarks to the Author):

This manuscript delineates a signaling pathway under the control of autophagic turnover of the scaffold protein TRAF3 that promotes Ras transformation. The authors propose that NDP52 mediates the selective degradation of TRAF3 in Ras mutant cells, which results in the activation of the NFKB family member RELB. However, RELB serves a NFKB independent function that suppresses SMAD4 transcription, thereby attenuating TGF-beta target gene expression. The authors propose autophagy inhibition suppresses Ras transformation and proliferation *in vivo* by augmenting TGF-beta pathway mediated suppression of proliferation; as a result, the RNAi against SMAD4 is able to restore tumor growth by Ras mutant autophagy-deficient cells.

The findings are interesting and the experiments are mostly well done; moreover, the paper provides a new perspective into how autophagy inhibition impairs the growth of Ras mutant cancer cells. However, the CRISPR deletion experiments require some additional technical and experimental controls and further studies are needed to more conclusively establish the importance of NDP52 mediated selective autophagy in the upstream control of TGF beta transcription. Finally, the authors should discuss their findings with regard to previous work linking autophagy to the control of metabolism and EMT during Ras driven tumorigenesis.

1) In Figure 1, the authors describe that CRISPR deletion of ATG5 has minimal effects *in vitro* but results in reduced tumor growth *in vivo*. The *in vitro* studies contrast with previous studies as well as the author's data in Supplemental Figure 1 using RNAi against ATGs and suggest that the phenotype of the CRISPR clones may be at least partly due to adaptation/clone selection in response to the complete genetic loss of ATG5. The authors provide one control in Figure 1D, but additional care should be taken in characterizing these reagents. First, the authors have used a single sgRNA for ATG5 deletion; preferably, an independent sgRNA to delete ATG5 should be used to generate an independent clone. At the very least, it is critical that sequencing evidence confirming distinct deletions in the ATG5 and ATG5-2 clones should be provided to demonstrate that the two CRISPR clones are truly independent rather than identical "siblings" during CRISPR knockout generation. Second, the *in vivo* tumor growth assays in Figure 1D-E must be conducted with two independent ATG5 clones to insure this phenotype is not a clonal artifact.

2) Virtually all of the analysis has been done in the context of ATG5 deletion so it is unclear whether these phenotypes are truly due to a general defect in autophagy versus a unique phenotype associated with loss of ATG5. The authors have provided some limited evidence in Figure 3 that TRAF3 is under the control of autophagy via the analysis of FIP200 RNAi but these studies should be extended further. It is most essential that the *in vivo* tumor growth assays in Figure 1 and TGFβ gene expression analysis in Figure 5 be repeated following the deletion/knockdown of a second independent ATG.

3) The authors emphasize the importance of selective autophagy but do not provide strong evidence in support of this idea. The role of NDP52 as the mediator of selective TRAF3 degradation is intriguing but preliminary. Additional work is needed to more robustly support the conclusion that NDP52 mediated degradation of TRAF3 is functionally responsible for the phenotypes that have been described. The authors should further analyze whether NDP52 loss-of-function phenocopies the effects of ATG5 deletion on Ras driven tumor growth and TGFβ pathway transcription to strengthen the cause-effect relationships between selective autophagy, control of the TGFβ pathway and Ras transformation.

4) Does TRAF3 accumulate in the ATG5 deletion xenografts *in vivo*?

5) To date, most studies of autophagy deficiency in Ras transformation have been linked to changes in metabolism including mitochondrial function, glutamine metabolism and glycolysis.

Although detailed metabolic analysis is beyond the scope of this paper, can the authors speculate whether the changes in metabolism observed by others are impacted by the TGF-beta/SMAD4 pathway or if this a completely different pathway working independently of metabolism?

6) Notably, in Ras transformed epithelial cells, classic studies from Julian Downward, Hartmut Beug and others (see Janda et al. JCB 2002 as one example) have demonstrated that TGF-beta serves as a potent and critical inducer of the pro-tumorigenic EMT program rather than as a mechanism of proliferative suppression, which the authors propose here. Moreover, studies have more recently demonstrated that Ras-transformed mammary cells exhibit a partial reversion of EMT following autophagy inhibition (Lock et al, Cancer Discovery, 2015), which one would not necessarily associate with enhanced TGF-beta signaling. This previous literature should be cited and the differences in this study regarding autophagy and TGFbeta in Ras transformed cells should be further discussed. Are TGF-beta associated EMT target genes enhanced upon autophagy inhibition and TRAF3/RELB activation in this model?

Reviewer #2 (Remarks to the Author):

Newman et al. (Wilkinson) Nature Communications

This is a complex manuscript in which the authors propose a functional connection between autophagy, ATG5 and tumorigenesis in a cancer cell model cell line expressing mutant Ras, as well as a functional and molecular connection between ATG5 in autophagy, TRAF3, RELB and repression of TGF-b-induced gene expression through Smad complexes.

I am reasonably familiar with autophagy, ATG5, TRAF3 and RELB, more familiar with Ras signaling, and very familiar with TGF-b/Smad signaling. Hence, and considering the complexity of the manuscript, I decided, after going through the manuscript a few times, to focus on what I perceived to be the core aspect of this manuscript, i.e. RELB-mediated repression of TGF-b-induced Smad signaling. My critique is shown below. Considering the substantial ground that the authors want to cover and the many conclusions they draw, I believe that the non-TGF-b/Smad aspects are also subject to substantial critique, but I leave this to the other reviewers. Overall, though, the manuscript comes over as highly complex, insufficiently worked out, and with insufficient basis for the conclusions and scenario proposed. The authors just want to conclude too much without sufficient basis.

As to the TGF-b/Smad signaling aspect and the RELB-mediated repression of TGF-b-induced Smad signaling, there is not enough basis for the conclusions drawn and the authors do not provide mechanistic insight. Furthermore, it is apparent that the authors are insufficiently familiar with the role of TGF-b signaling in cancer and the underlying signaling mechanisms. Moreover, they may be misguided in their reading of the literature, or may misrepresent the current insights in this field that are relevant to their research. Below are some specifics related to TGF-b/Smad signaling and the repression by RELB.

- The authors work in the assumption that TGF-b signaling goes through Smad2-Smad4. This is not correct. In addition to the substantial level of TGF-b-induced non-Smad signaling, the Smad-mediated changes in gene expression are primarily mediated by receptor-activated Smad3 (albeit with some participation by Smad2) with Smad4 as coactivator. The authors do not address whether RELB interacts with Smad3, and actually never mention Smad3. Moreover the SBE reporter they use is a Smad3 reporter, yet they evaluate Smad2 and Smad4. Additionally, Smad2 does not bind DNA directly, but can occupy DNA through recruitment by Smad3 or Smad4. So, they really do not evaluate correctly the TGF-b-induced, Smad-mediated changes in gene expression.

- In the end, they conclude that RELB represses TGF-b/Smad-induced gene expression through binding to Smad4. However, (1) Smad3 was not considered, as pointed out above, (2) Smad2 was

not really evaluated, (3) that RELB binds Smad4 is not a mechanism, especially considering that currently >200 transcription factors are known to bind Smads. A mechanism of repression by RELB needs to be presented, yet is not proposed.

- The authors make wrong statements about TGF- β signaling in cancer that they then use as their basis for their interpretation (and conclusions) of their results. (1) "The tumor suppressive TGF- β -Smad4 pathway" (Summary) is a wrong statement. Yes, Smad4 can act as tumor suppressor in pancreas and colon carcinomas but Smad4 is a Smad coactivator involved in both the tumor suppressor as well as tumor promoting activities of TGF- β . (2) "RAS mutant cancer cells commonly exhibit decreased sensitivity to TGF- β " (in Introduction) is wrong. These cells show decreased sensitivity to TGF- β -induced growth inhibition, but not to many/most other responses, including gene responses. (3) "TGF- β -driven transcription is anti-tumorigenic" (page 28) is wrong. There is a tumor suppressor component related to growth inhibition, but it also promotes tumor progression. (4) Following this statement the authors conclude that "Thus, the suppression of TGF- β transcriptional output by autophagy and RELB/NF- κ B is a component of the pro-tumorigenic effect of autophagy". This conclusion, the major conclusion of this report, is based on the previous statement and is therefore not scientifically justified.

- Additional misrepresentations: (1) ALKi (the Tbr1/ALK-5 inhibitor) inhibits upstream signaling that culminates in phosphorylation of SMAD2 (P-SMAD2). This is only partially true since Smad3 is the major effector (see above). (2) The authors present ALKi as an inhibitor of ALK-5 tyrosine kinase. This is wrong as Tbr1/ALK-5 phosphorylates primarily on Ser and Thr, and the inhibitor also inhibits ALK-2 and ALK-4. (3) (page 18) "A classic TGF- β -induced functional response was observed upon RELB signaling, namely abundance of paxillin adhesion foci". I know the field very well, and I have never heard about this response (nor is a reference given); so it is hard to believe that this is a "classic TGF- β response". (4) (page 22, 1st paragraph) The authors state that H3K4 dimethylation is upregulated at the promoters evaluated, and therefore conclude that this indicates elevated promoter activity. In contrast, H3K4 dimethylation commonly correlates with gene repression. (5) (page 25, toward end) "When Smad4 activation by cytosolic TGF signaling was blocked by ALKi" does not make sense since TGF- β does not activate Smad4.

- The authors state that they found bona fide TGF- β /Smad target genes to be repressed by RELB. However, first of all, I do not know on what basis they conclude that these are bona fide target genes, since many of these are not known to be direct TGF- β /Smad target genes. Second, Smads act through cooperation with other transcription factors, often AP-1 transcription factors, and I know that some of the identified targets are regulated through cooperation of Smad3/4 with AP-1. This is important since AP-1 signaling is expected to be activated as a result of the mutant Ras expression. Yet the authors do not consider the possibility that AP-1, or any transcription factors downstream from Ras, are involved in this control by autophagy and ATG5.

Reviewer #3 (Remarks to the Author):

Newman et al have investigated the effect of RAS-dependent autophagy in transcription regulation in A549 lung adenocarcinoma cells, in order to identify novel mechanisms that mediate autophagy-dependent cell growth and tumorigenesis by oncogenic RAS. They have identified an autophagy-dependent upregulation of TGF β -dependent gene transcription which was attributed to degradation of TRAF3 and activation of RELB. RELB was shown to antagonize SMAD-dependent gene transcription in a manner that is independent from its consensus NF- κ B site-binding capacity and depends on its ability to interact with SMAD proteins. The study introduces novel aspects of RAS-dependent signalphagy but lacks sufficient data to establish the biological relevance and generalization of its conclusions. The authors must address the following issues.

1. The relevance of TRAF3 downregulation and RELB activation in lung adenocarcinoma cell growth in vivo must be established by evaluating the effect of their respective upregulation and downregulation on the growth of WT A549. In addition, the effect of TRAF3 downregulation and RELB activation on the in vivo growth of DeltaATG5 A549 cells must be evaluated.

2. In order to exclude cell line dependent effects key experiments including the evaluation of in vivo growth of lung adenocarcinoma cell lines must be repeated with at least one more lung adenocarcinoma cell line.
3. To establish the dependence of TRAF3 degradation and RELB activation on oncogenic RAS signaling the authors must use a cell line model with inducible expression of an oncogenic RAS mutant and demonstrate that in such cells the changes in TRAF3 and RELB can be induced by the expression of oncogenic RAS.
4. Inactivation of ATG5 by CRISPR/Cas in A549 does not affect their growth in vitro whereas downregulation of ATG5 by RNA interference inhibits their growth in vitro. What is the reason for this discrepancy? Furthermore the fact that in both cases TGFbeta-pathway target genes are upregulated, questions the importance of TGFbeta-pathway dependent gene expression upregulation for the growth of A549 in vitro and underscores the need for additional experiments to generalize the importance of RAS and autophagy driven upregulation of SMAD-dependent transcription in the growth of lung adenocarcinoma cells in vivo.
5. The authors should determine whether the effect of RAS-driven autophagy on TRAF3 is specific or whether other inducers of autophagy also target TRAF3 for autophagic degradation.
6. The authors have shown previously that TBK1-dependent autophagy in A549 cells leads to RELB nuclear localization (Newman et al, PLoS One, 2012). How is TBK1 activity linked to TRAF3 downregulation? TRAF3 has been shown to mediate TBK1 activation in innate immune signaling. Is there a different mechanism that links the two molecules in RAS-driven autophagy? The authors should address these points.
7. In figures 2C, 2D, 2E, 4C, 4E, 5A, 6B, 6C, 6E the statistical significance of the observed differences must be determined.
8. On page 4 it is incorrectly stated that TGFbeta ligates receptor tyrosine kinases instead of receptor serine/threonine kinases.

Response to reviewers

Firstly, we would like to thank the reviewers for their comments. We feel that in addressing these we have produced a more robust and well-rounded manuscript (enclosed). We have provided a number of additional experiments and controls. We discuss these changes on a point-by-point basis below, in the context of the individual reviewer's comments. We have also taken on board other comments in order to make some modifications to the text for clarity and for scope of discussion.

Reviewer 1

1) In Figure 1, the authors describe that CRISPR deletion of *ATG5* has minimal effects in vitro but results in reduced tumor growth in vivo. The in vitro studies contrast with previous studies as well as the author's data in Supplemental Figure 1 using RNAi against *ATGs* and suggest that the phenotype of the CRISPR clones may be at least partly due to adaptation/clone selection in response to the complete genetic loss of *ATG5*.

It was surprising to us, initially, that we could make *ATG5* null clones. However, it is notable that *ATG5* knockdown by siRNA only leads to reduced cell-cycle progression, not cell death, over 72 h. This point was not explicitly made in the original manuscript. We now make this clearer on p6 (top para) and provide new time-lapse movies to underscore this point, relating to Supplemental Figure 1 (Supp Movie 1-3). Thus, while clonal selection/adaptive pressure might play a role in determining the characteristics of the CRISPR clone reagents, adaptive pressure may not be as high as it ostensibly appears. Also, in response to the request for new knockouts of other *ATGs*, these are done with population level CRISPR rather than individual clones (see points below), at least diminishing concerns of clonal selection.

Furthermore, key phenotypes such as TRAF3 stabilisation, RELB and characteristic transcriptional changes are seen upon both acute RNAi and complete genetic loss of multiple *ATG* genes and/or *NDP52* (see points below for details, along with references to Figures).

The authors provide one control in Figure Fig 1D, but additional care should be taken in characterizing these reagents. First, the authors have used a single sgRNA for *ATG5* deletion; preferably, an independent sgRNA to delete *ATG5* should be used to generate an independent clone. At the very least, it is critical that sequencing evidence confirming distinct deletions in the *ATG5* and *ATG5-2* clones should be provided to demonstrate that the two CRISPR clones are truly independent rather than identical "siblings" during CRISPR knockout generation.

We agree - sequencing evidence is now provided in new Fig. 1B.

Second, the in vivo tumor growth assays in Fig 1D-E must be conducted with two independent *ATG5* clones to insure this phenotype is not a clonal artifact.

New data is now provided in Fig 1F. Again, the second independent Δ *ATG5* clone fails to form sizeable tumors. Also see Fig 4E which was generated in response to the next point made by the reviewer – in this experiment pooled (i.e. not clonal) CRISPR targeting of *FIP200* and *NDP52* also abrogated tumorigenesis.

2) Virtually all of the analysis has been done in the context of *ATG5* deletion so it is unclear whether these phenotypes are truly due to a general defect in autophagy versus a unique phenotype associated with loss of *ATG5*. The authors have provided some limited evidence in Figure 3 that TRAF3 is under the control of autophagy via the analysis of *FIP200* RNAi but these studies should be

extended further. It is most essential that the *in vivo* tumor growth assays in Figure 1 and TGFβ gene expression analysis in Figure 5 be repeated following the deletion/knockdown of a second independent ATG.

We have now performed a number of additional experiments to corroborate our initial findings:

We now provide new data showing TRAF3 stabilisation on *ULK1* knockdown (Fig S5C-E), transcriptional upregulation similar to *ATG5* knockdown/deletion upon *ULK1* knockdown (new Fig. 2D), new data showing TRAF3 stabilisation on pooled (i.e. not clonal) CRISPR of *FIP200* and *NDP52* in A549 cells (Fig. 4D and 3G, respectively), and cognate upregulation of TGFβ genes (Supp. Fig. 5F).

We also now perform pooled (not clonal) CRISPR deletion of *ATG5* in a second human cell lines (NCI-H23) and show upregulation of TRAF3 protein and TGFβ-driven transcripts (Figs. 4F and Supp. Fig. 6A).

FIP200 and *NDP52* knockout also inhibit tumorigenesis (Fig. 4E) and, importantly, the effect of *FIP200* knockout in inhibiting tumor growth is reversed by *SMAD4* knockdown, as previously established for *ATG5* (Figs. 8D-E).

3) The authors emphasize the importance of selective autophagy but do not provide strong evidence in support of this idea. The role of *NDP52* as the mediator of selective TRAF3 degradation is intriguing but preliminary. Additional work is needed to more robustly support the conclusion that *NDP52* mediated degradation of TRAF3 is functionally responsible for the phenotypes that have been described. The authors should further analyze whether *NDP52* loss-of-function phenocopies the effects of *ATG5* deletion on Ras driven tumor growth and TGFβ pathway transcription to strengthen the cause-effect relationships between selective autophagy, control of the TGFβ pathway and Ras transformation.

As outlined in response to point 2) we have now performed CRISPR/Cas9 for *NDP52* in addition to other *ATG* genes and demonstrated upregulation of TGFβ pathway transcription and loss of tumorigenicity.

4) Does TRAF3 accumulate in the *ATG5* deletion xenografts *in vivo*?

Due to the very limited growth of *ATG5*-deleted xenografts we were unable to extract enough material for both immunohistochemistry and meaningful immunoblot analysis. Quantitative TRAF3 immunohistochemistry is also not possible as an alternative due to lack of specificity of all available antibodies in this application.

This notwithstanding, it is worth noting that either of the likely alternate findings of a) maintenance of TRAF3 upregulation and consequent slow proliferation of tumor cells *in vivo*, or b) reversal of the increased TRAF3 levels during hypothetical selective pressure for tumorigenicity, could be reconciled with our current findings.

5) To date, most studies of autophagy deficiency in Ras transformation have been linked to changes in metabolism including mitochondrial function, glutamine metabolism and glycolysis. Although detailed metabolic analysis is beyond the scope of this paper, can the authors speculate whether the

changes in metabolism observed by others are impacted by the TGF-beta/SMAD4 pathway or if this is a completely different pathway working independently of metabolism?

We wholly agree. Both possibilities remain open at the moment and we have highlighted this concept in the Discussion section (final para., p18)

6) Notably, in Ras transformed epithelial cells, classic studies from Julian Downward, Hartmut Beug and others (see Janda et al. JCB 2002 as one example) have demonstrated that TGF-beta serves as a potent and critical inducer of the pro-tumorigenic EMT program rather than as a mechanism of proliferative suppression, which the authors propose here. Moreover, studies have more recently demonstrated that Ras-transformed mammary cells exhibit a partial reversion of EMT following autophagy inhibition (Lock et al, Cancer Discovery, 2015), which one would not necessarily associate with enhanced TGF-beta signaling. This previous literature should be cited and the differences in this study regarding autophagy and TGFbeta in Ras transformed cells should be further discussed. Are TGF-beta associated EMT target genes enhanced upon autophagy inhibition and TRAF3/RELB activation in this model?

We agree completely with the reviewer. Indeed, in the original manuscript a decision was taken to focus on the capability of TGFβ to repress tumor growth, at least in some contexts, simply to streamline the narrative (although we did acknowledge the “double-edged” sword of TGFβ signaling in the Discussion).

TGFβ has been established in the literature as having a pro-EMT effect on A549 cells *in vitro*. We would not argue that autophagy has no effect on EMT target genes. For instance, some well-known EMT target genes (*CDH2* and *SNAI2*, i.e. N-cadherin and Slug) are indeed enhanced upon autophagy inhibition and RELB inhibition in our model system (e.g. Fig. S7). However, these are not ranked in the top fold changes for either inhibitory regimen and so do not appear in the main figures.

The outcome of TGFβ stimulation is an integration of SMAD signaling and other, non-canonical pathways, several of which (MAP kinases, RHO GTPases etc.) impinge on EMT and motility. Here, we only propose autophagy inhibition to affect SMAD-driven gene transcription (and even then probably with varying magnitudes of effect on different promoters relative to direct TGFβ stimulation, although addressing this is outwith the scope of the manuscript).

The literature regarding autophagy and EMT is complex. Several recent manuscripts have demonstrated a capacity for autophagy to repress EMT via yet different mechanisms from regulation of SMAD output, in various cellular contexts (e.g. Grassi et al, Cell Death Dis 6:e1880, Qiang et al PNAS 111:9241). On the other hand, teams such as Lock et al have clearly shown a role for autophagy in promoting aspects of EMT (in this example via pro-EMT factor secretion). We consider selective autophagy to be a collection of mechanistically discrete processes, the potential for all such processes (as well as bulk autophagy) becoming abrogated when *ATG* genes are targeted. We consider it likely that the direction and magnitude of effect of autophagy on EMT is likely to be the integrated, net effect of the differing levels of engagement of all of these various autophagy events in a particular cellular context.

We concisely bring together the above concepts in new text, citing the relevant literature (beginning p16, bottom paragraph).

Reviewer 2

I am reasonably familiar with autophagy, ATG5, TRAF3 and RELB, more familiar with Ras signaling, and very familiar with TGF- β /Smad signaling. Hence, and considering the complexity of the manuscript, I decided, after going through the manuscript a few times, to focus on what I perceived to be the core aspect of this manuscript, i.e. RELB-mediated repression of TGF- β -induced Smad signaling.

The RELB-dependent repression of SMAD target gene expression is an important aspect of the manuscript but not *the* solitary, “core” aspect.

The following responses to individual reviewer points should clarify further what we do and, as importantly, what we do not endeavour to claim in regard of this section of the manuscript.

A) The authors work in the assumption that TGF- β signaling goes through Smad2-Smad4.

To clarify, we do not work from the assumption that TGF β signaling goes “through” SMAD2/4, if direct activation of SMAD2/4 by post-translational modification is what is meant here.

Unequivocally, SMAD2/4 are recruited to promoters analysed (shown empirically [new Supp. Fig. S9 also extends the data to SMAD2 now], and also as one would expect as part of a multi-SMAD complex).

We show that SMAD4 and SMAD2 are required for recruitment of RELB to promoters (previous Figs. and new Supp. Fig. S9).

RELB has known transcriptional inhibitory activity. However, we do not claim this activity is mediated directly via targeting of individual trans-activatory activity of SMAD2 or SMAD3 or SMAD4 (or indeed any of myriad of other TFs, coactivators, corepressors etc. that might be present at promoters in question). However, without recruitment to chromatin by binding SMAD(s) RELB is unlikely to effect changes in gene transcription. SMAD2 and 4 are required for this.

B) In addition to the substantial level of TGF- β -induced non-Smad signaling, the Smad-mediated changes in gene expression are primarily mediated by receptor-activated Smad3 (albeit with some participation by Smad2) with Smad4 as coactivator.

We do not say that SMAD2 is activated by the receptor signaling, if by this stimulation of a hypothetical transactivation activity is intended. However, SMAD2 *is* recruited to promoters analysed, as is SMAD4 (which does support gene activation more directly). This is stimulated upon TGF β treatment or is repressed by ALKi. This is shown empirically in some control experiments (e.g. Supp. Fig S9) but is also wholly in line with current models.

The important point is that these SMADs interact with RELB and inhibiting the recruitment of these SMADs to promoters will affect RELB recruitment. We do not claim that RELB directly inhibits SMAD2 or SMAD4 “activity” *per se*. Nonetheless, recruited RELB inhibits gene transcription.

We did originally describe the activity of RELB as an antagonism of SMAD4 activity, which it clearly is in functional terms. However, one could as easily frame this as antagonism of SMAD complexes in such functional terms. This is perhaps now more appropriate, as we have extended some of the interaction and CHIP data to other SMADs (see points below). It may also avoid the impression that we are suggesting RELB somehow does something directly to SMAD4 molecules to block their

contribution to gene activation. We have accordingly modified the title, and also the model (Fig. 9), to refer to antagonism of SMAD(s)/SMAD complexes rather than SMAD4 specifically.

C) The authors do not address whether RELB interacts with Smad3, and actually never mention Smad3.

For the reasons outlined above in response to points A) and B), this criticism does not diminish our conclusions that RELB is recruited to genes via members of the multi-SMAD complex (SMAD2 and SMAD4), thus likely facilitating its inhibition of gene transcription from these genomic locales.

Nonetheless, we now show that RELB interacts with SMAD3 (Fig. 7A).

We also now show SMAD3 phosphorylation is unperturbed by ATG/RELB inhibition (Figs. 2G, 5G)

D) Moreover the SBE reporter they use is a Smad3 reporter, yet they evaluate Smad2 and Smad4. Additionally, Smad2 does not bind DNA directly, but can occupy DNA through recruitment by Smad3 or Smad4. So, they really do not evaluate correctly the TGF- β -induced, Smad-mediated changes in gene expression.

We do not say that this promoter binds SMAD2 or SMAD4 directly (page 13, upper paragraph). We say that heteromeric SMAD assemblies bind these elements (which includes direct and indirect interactions)(e.g. Introduction, bottom p4). SMAD2 and SMAD4 co-transfection, as would be expected, can be seen empirically in our data to activate this SBE, by driving the intracellular equilibrium toward further occupancy of the reporter by multi-SMAD complexes (these will contain endogenous proteins such as SMAD3 and other TF/co-activators/co-repressors). We do not suggest the reporter is reporting SMAD3 alone nor, conversely, SMAD2/4 alone.

The important point for this setup is that it allows analysis of the effects of RELB on active SBE reporter when we know SMADs to which RELB binds, e.g. SMAD2/4 consistent with our initial binding assays, are resident there.

- In the end, they conclude that RELB represses TGF- β /Smad-induced gene expression through binding to Smad4.

We conclude that RELB both represses transcription from - and is recruited to - SMAD-complex occupied promoters, by binding to SMAD4 (and now also SMAD2, see new Figure S9). This binding may be direct or indirect, and certainly does not preclude a role for other SMADs or TFs. However, SMAD4 and SMAD2 are clearly required for recruitment.

We do not suggest that RELB inhibits SMAD4 directly and have carefully re-evaluated our phrasing throughout the manuscript, to ensure that this is not implied. The precise nature of RELB-mediated transcriptional repression of promoters remains to be determined (a long-standing and difficult issue in the NF- κ B field, even where RELB represses canonical NF- κ B binding promoters).

However, (1) Smad3 was not considered, as pointed out above

We have addressed this point extensively, in the collective response to several other individual criticisms (above).

(2) Smad2 was not really evaluated.

We now show that SMAD2 is recruited to test promoters by TGF β stimulation and, importantly, is required for RELB recruitment to promoters (Supp. Fig S9). This mirrors the SMAD4 findings, corroborating the recruitment of RELB to SMAD complexes on chromatin.

(3) that RELB binds Smad4 is not a mechanism, especially considering that currently >200 transcription factors are known to bind Smads. A mechanism of repression by RELB needs to be presented, yet is not proposed.

RELB does not just merely bind SMAD4 (and SMAD2/3) but is recruited to promoters with and dependent upon SMAD complexes, as outlined above.

The referee is indeed correct that this does not provide a complete molecular mechanism for *how* RELB can repress transcription when recruited to genes. This is a long-standing and multi-faceted question in the NF- κ B field and we do not attempt to resolve it here.

We now make especially clear in the manuscript, throughout, that antagonism of SMAD function is at the level of transcriptional output from the promoters bound to both SMAD(s) and RELB, rather than some direct interaction of RELB to inhibit a particular function of a particular SMAD protein.

We think these findings are proportionate within the scope and breadth of the study (which we perceive as wider than perhaps the referee does, see initial point in this response).

4) The authors make wrong statements about TGF-b signaling in cancer that they then use as their basis for their interpretation (and conclusions) of their results. (1) "The tumor suppressive TGF-b-Smad4 pathway" (Summary) is a wrong statement. Yes, Smad4 can act as tumor suppressor in pancreas and colon carcinomas but Smad4 is a Smad coactivator involved in both the tumor suppressor as well as tumor promoting activities of TGF-b.

We are aware of this. The manuscript was written for simplicity, to provide a clear exposition of the core arguments and may have admittedly been too concise. We had indeed referred to tumor promoting activities of TGF β in the original Discussion. However, we expand upon these now (including EMT) in the Introduction (bottom page 4, onward) and at greater length in the discussion (bottom page 16, onward).

(5) "RAS mutant cancer cells commonly exhibit decreased sensitivity to TGF-b" (in Introduction) is wrong. These cells show decreased sensitivity to TGF-b-induced growth inhibition, but not to many/most other responses, including gene responses.

This sentence followed a prior statement in the manuscript, discussed in the point above, regarding the anti-tumorigenic effects of TGF β . So, taken in context, it was meant to reflect desensitisation to anti-tumorigenic effects. We have clarified this now in the revised text (bottom page 4, onward).

(6) "TGF-b-driven transcription is anti-tumorigenic" (page 28) is wrong. There is a tumor suppressor component related to growth inhibition, but it also promotes tumor progression.

This statement again reflects the idea the ground covered in response to points 4 and 5 above. We no longer formulate this statement in this manner, and, as described specifically above, endeavour

to make clear that TGF β and SMAD complexes play a multifaceted, context-dependent role in tumorigenesis (page 14, bottom paragraph, now refers to both pro- and anti-tumorigenic roles).

(7) Following this statement the authors conclude that “Thus, the suppression of TGF- β transcriptional output by autophagy and RELB/NF- κ B is a component of the pro-tumorigenic effect of autophagy”. This conclusion, the major conclusion of this report, is based on the previous statement and is therefore not scientifically justified.

Please see points 4-6.

Also, this conclusion is supported by the empirical observations in this manuscript that a) autophagy inhibition (and RELB inhibition) enhances TGF β transcriptional output and b) that SMAD4 knockdown promotes tumorigenesis in this context.

- Additional misrepresentations:

(1) ALKi (the TbRI/ALK-5 inhibitor) inhibits upstream signaling that culminates in phosphorylation of SMAD2 (P-SMAD2). This is only partially true since Smad3 is the major effector (see above).

In the original context, this statement was formulated to explain use of phosphorylation of SMAD2 as a control readout for the integrity of the inhibitor (i.e. merely to show the batch in use and conditions of use indeed work on receptors). It was not a misrepresentation.

In any case, we have modified for clarity to “cytosolic signaling events... can be detected by... phosphorylation of SMAD2 and SMAD3” (middle page 8).

(2) The authors present ALKi as an inhibitor of ALK-5 tyrosine kinase. This is wrong as TbRI/ALK-5 phosphorylates primarily on Ser and Thr, and the inhibitor also inhibits ALK-2 and ALK-4.

We thank the reviewer for their diligence in detecting the erroneous description of ALK5 as an RTK rather than RSTK. This was a genuine oversight in editing of the manuscript, and has been corrected.

The salient point in context is that ALKi ultimately modulates SMAD occupancy of chromatin (as shown empirically in the manuscript). This is what it is employed to do as one of a portfolio of overlapping tools (e.g. TGF β treatment, SMAD knockdown etc.) that all achieve this end.

We have now edited the text to refer to it as an ALK2/4/5 inhibitor, acknowledging that it also targets some other TGF β and TGF β -related ligand receptors. However, this does not undermine its use in the context of this study, as described in the paragraph above (middle page 8, also in Methods).

(3) (page 18) “A classic TGF- β -induced functional response was observed upon RELB signaling, namely abundance of paxillin adhesion foci”. I know the field very well, and I have never heard about this response (nor is a reference given); so it is hard to believe that this is a “classic TGF- β response”.

TGF β will drive the concentration of focal adhesion proteins in a number of cell lines (including A549). We have detected this here with paxillin staining, as this is readily quantifiable by image analysis in A549 cells. We have rephrased to avoid the use of the word “classic” – this was merely meant to indicate a long-standing observation we are aware of from the cytoskeleton field - and provided a reference for this (top page 12).

(4) (page 22, 1st paragraph) The authors state that H3K4 dimethylation is upregulated at the promoters evaluated, and therefore conclude that this indicates elevated promoter activity. In contrast, H3K4 dimethylation commonly correlates with gene repression.

We do not agree that this is the current evaluation of H3K4 di-methylation. Di- and Tri- at H3K4 commonly correlate with active gene transcriptional units when present proximal to genes (i.e. close to TSS) or intragenically. There are isolated exceptions to this rule, as there are for many chromatin marks, but this is the general consensus in the literature. We cite a prominent review article in this regard in the paper (page 12, third paragraph).

(5) (page 25, toward end) “When Smad4 activation by cytosolic TGF signaling was blocked by ALKi” does not make sense since TGF-b does not activate Smad4.

In context, this statement refers to regulation of chromatin occupancy by SMAD4 (it immediately preceded a reference to a Figure where this was the readout). Chromatin occupancy is indeed stimulated by TGF β . This is consequent from initial activation of R-SMAD(s), but the end result is recruitment of SMAD4.

We agree that activation was a suboptimal choice of term here. We have rephrased to avoid unintentional misunderstanding here i.e. replacing “activation” with stimulation of “SMAD4 DNA-binding” (again, we know this is indirect) (page 14, upper paragraph).

The authors state that they found bona fide TGF-b/Smad target genes to be repressed by RELB. However, first of all, I do not know on what basis they conclude that these are bona fide target genes, since many of these are not known to be direct TGF-b/Smad target genes.

An extensive list of justification of the genes taken from the array, and reference to their status as downstream of TGF β , i.e. “TGF β -driven” genes (direct or indirect) and/or as validated SMAD2/3/4 binding genes is provided in the manuscript (Supp. Table S2, Supp. Table S4, Supplemental References).

Empirical validation of TGF β -driven status and/or SMAD-binding status is also performed throughout the manuscript or in Supplementary materials for many genes, as carefully laid out in the legends and content of Supp. Tables S2 and S4.

We have also now been through the manuscript very carefully to ensure the precise, differentiating terms TGF β -driven and TGF β /SMAD target genes are clearly defined and are used consistently. This is because, clearly, not all genes upregulated by TGF β are direct SMAD target genes. A minority will be altered secondarily to SMAD driven changes in transcriptional profile and cellular biology. A yet further minority will be activated relatively directly by TGF β upstream signaling, but via non-canonical mechanisms. The point is that knockdown or knockout of ATG5/RELB leads to a global transcriptional profile resembling TGF β stimulation and that honing in on a (relatively large) number

of direct i.e. SMAD-binding target genes (again, extensively justified in the main and supplementary materials, above), we show that these are repressed by RELB.

Second, Smads act through cooperation with other transcription factors, often AP-1 transcription factors, and I know that some of the identified targets are regulated through cooperation of Smad3/4 with AP-1. This is important since AP-1 signaling is expected to be activated as a result of the mutant Ras expression. Yet the authors do not consider the possibility that AP-1, or any transcription factors downstream from Ras, are involved in this control by autophagy and ATG5.

Yes, we agree, it is entirely possible that AP-1 factors are involved in the control of transcription by autophagy. However, it is not justifiable to use this possibility to negate the findings of this paper on the role of RELB. The substantial findings are and remain (as outlined above) that RELB represses this cohort of genes, can bind SMAD and is recruited to promoters of SMAD-binding genes via this interaction. Regardless of AP-1 involvement within such a pathway, it is clear that RELB counteracts SMAD-dependent transcriptional output (although as acknowledged above a detailed study of how RELB counteracts gene expression e.g. recruitment co-repressors etc. to these loci, is outwith the scope of this paper).

Reviewer 3

Newman et al have investigated the effect of RAS-dependent autophagy in transcription regulation in A549 lung adenocarcinoma cells, in order to identify novel mechanisms that mediate autophagy-dependent cell growth and tumorigenesis by oncogenic RAS. They have identified an autophagy-dependent upregulation of TGFbeta-dependent gene transcription which was attributed to degradation of TRAF3 and activation of RELB. RELB was shown to antagonize SMAD-dependent gene transcription in a manner that is independent from its consensus NF-kappaB site-binding capacity and depends on its ability to interact with SMAD proteins. The study introduces novel aspects of RAS-dependent signalphagy but lacks sufficient data to establish the biological relevance and generalization of its conclusions.

We provide substantial extra data as requested (see point-by-point below). In addition to the data requested here, please also see the extensive extra data on FIP200, ULK1 and NDP52 function, both *in vitro* and *in vivo*, provided in response to Reviewer 1's queries, which further corroborate our findings.

1. The relevance of TRAF3 downregulation and RELB activation in lung adenocarcinoma cell growth *in vivo* must be established by evaluating the effect of their respective upregulation and downregulation on the growth of WT A549.

RELB deletion with two separate CRISPR/Cas9 gRNA in pooled cultures is sufficient to abrogate tumorigenesis, revealing the requirement of this pathway downstream of autophagy. This is shown in new data Figs. 5A, B.

In addition, the effect of TRAF3 downregulation and RELB activation on the *in vivo* growth of DeltaATG5 A549 cells must be evaluated

TRAF3 knockdown is sufficient to regain growth of Δ ATG5 A549 cells over the first 30 days *in vivo* (A, below). However, long term these tumors do regress. Intriguingly, TRAF3 knockout *per se* dramatically compromises the growth of parental (autophagy-competent) A549 cells (B, below). Taking these data together, along with the above finding that RELB is required for tumorigenesis, this suggests that TRAF3 *per se* is required at some level for tumorigenesis. However, de-regulated, high levels (which are clearly causative of RELB inhibition, as seen when autophagy is inhibited) can compromise tumorigenesis.

At least currently, we have not included these data in the manuscript, due to the already complex nature of the manuscript after inclusion of other revisions.

2. In order to exclude cell line dependent effects key experiments including the evaluation of *in vivo* growth of lung adenocarcinoma cell lines must be repeated with at least one more lung adenocarcinoma cell line.

We now show that NCI-H23 cells accumulate TRAF3 protein when ATG5 is targeted by CRISPR and exhibit inhibition of NF κ B2 processing (Fig. 4F). This phenocopies A549 cells. As with A549 cells NCI-H23 also exhibit upregulation of mRNA levels of TGF β targets (Supp. Fig. S6A.)

These effects were also shown in oncogenic RAS-transformed MEFs, e.g. Fig. 2F, Fig 4G, H).

We attempted to xenograft the NCI-H23 model, but all lines failed to established tumors. This precluded *in vivo* analysis. However, please see response to point 4, below, where we can assuage concerns about the effect of autophagy inhibition *in vitro* versus *in vivo*.

3. To establish the dependence of TRAF3 degradation and RELB activation on oncogenic RAS signaling the authors must use a cell line model with inducible expression of an oncogenic RAS mutant and demonstrate that in such cells the changes in TRAF3 and RELB can be induced by the expression of oncogenic RAS.

We have performed expression of oncogenic RAS in murine cells in Fig. 4G-H and have shown an increase in the rate of TRAF3 flux via the autophagy pathway and an induction of RELB, which is dependent upon autophagy (and is reversed by TRAF3 knockdown Fig. 4I). Oncogenic RAS is introduced by virus here with close to 100% efficiency – these are pools, not clones of cells with RAS expression, analysed when cells have recovered from selection to kill off the small percentage of non oncogenic-RAS expressing cells.

Importantly, we do not make any claims as to how acute the effect of RAS mutant introduction is on TRAF3, i.e. a consequence of immediate signaling via acute activation of MAP kinase / RalB etc. or a more delayed response (an inducible cell line would indeed be required for this). Although, see detailed discussion of TBK1 signaling, which occurs downstream of oncogenic RAS->RalB, below in response to point 6.

4. Inactivation of ATG5 by CRISPR/Cas in A549 does not affect their growth *in vitro* whereas downregulation of ATG5 by RNA interference inhibits their growth *in vitro*. What is the reason for

this discrepancy? Furthermore the fact that in both cases TGFbeta-pathway target genes are upregulated, questions the importance of TGFbeta-pathway dependent gene expression upregulation for the growth of A549 *in vitro* and underscores the need for additional experiments to generalize the importance of RAS and autophagy driven upregulation of SMAD-dependent transcription in the growth of lung adenocarcinoma cells *in vivo*.

There is reduction in cell-cycle progression upon knockdown of *ATG5* or *FIP200* by siRNA - but no cell death. This point was not explicitly made in the original manuscript. We now make this clearer on p6 (top para) and provide new time-lapse movies to underscore this point, relating to Supplemental Figure 1 (Supp Movie 1-3). We do not conclude that TGFβ/SMAD upregulation necessarily causes this growth defect. Naïve cells that have previously not experienced compromise of autophagy could be induced to cycle more slowly (at least temporarily) for any number of reasons - numerous possibilities exist within the literature and it is possible as some yet undiscovered mechanism could be at play here.

Notably, we have empirically tested the effect of continuous TGFβ treatment on A549 cells *in vitro* and, even with this direct, sustained and synchronous method of activating the pathway we see milder anti-proliferative effects than with *ATG5* knockdown, despite this being latter treatment being slower, less synchronous and much less effective on TGFβ-driven transcription (e.g. Fig. 2C).

Nonetheless, it is possible that there is a contribution of TGFβ/SMAD pathway upregulation upon autophagy inhibition on proliferation *in vitro*, but cells are still able to survive and grow out. In this regard, it is feasible that some adaptation occurs upon stable loss of autophagy function (i.e. selection for cells with only medium levels of TGFβ/SMAD activation). Nonetheless, new data show that *FIP200* deletion (even in pooled cells) is tolerable *in vitro* but tumor growth is wholly abrogated *in vivo*, again in a SMAD4-dependent manner (Fig 8D, E). We feel this data further corroborates the acquisition of SMAD-sensitivity *in vivo*, previously shown with *ATG5* deletion.

5. The authors should determine whether the effect of RAS-driven autophagy on TRAF3 is specific or whether other inducers of autophagy also target TRAF3 for autophagic degradation.

Indeed, we have been unable to convincingly detect EBSS driven turnover of TRAF3 in an *ATG5*-dependent manner, even with long-term amino acid starvation (Fig. S6D, discussed top of page 11).

6. The authors have shown previously that TBK1-dependent autophagy in A549 cells leads to RELB nuclear localization (Newman et al, PLoS One, 2012). How is TBK1 activity linked to TRAF3

downregulation? TRAF3 has been shown to mediate TBK1 activation in innate immune signaling. Is there a different mechanism that links the two molecules in RAS-driven autophagy? The authors should address these points.

This is a very interesting point. It is an attractive idea that there may be a link between TBK1-dependent autophagy and regulation of TRAF3 levels. This is not a link that we have been able to conclusively test and thus we have not elaborated upon in the manuscript itself. Firstly, new data (included here for reviewer only, at least currently) show that, in A549 cells, multiple TBK1 siRNAs (panels A-C, below) or inhibition of TBK1 activity with the inhibitor MRT67037 (panels D, E, below) (the methods used in Newman et al 2012) do indeed result in TRAF3 protein level increase. However, this is at least partially transcriptional (unlike the effects of ATG gene knockdown or knockout, which are shown throughout our current manuscript to be post-transcriptional). Furthermore, in non-RAS transformed MEFs (panel F, below), there is no effect of *ATG5* deletion (as also shown in our current manuscript) but treatment with the TBK inhibitor co-operates with *ATG5* deletion to increase TRAF3 protein levels. This suggests, again, at least one other route for TRAF3 regulation that is independent of Atg5, possibly transcriptional.

The complexity of the role of TBK1 make this data, we feel, unsuitable for inclusion in the current manuscript.

We have also checked whether the upregulated TBK1 activity first reported by Mathew et al in autophagy-deficient cells (Mol Cell, 2014) is dependent upon TRAF3 accumulation, and obtained no definitive evidence for this.

7. In figures 2C, 2D, 2E, 4C, 4E, 5A, 6B, 6C, 6E the statistical significance of the observed differences must be determined.

Thank you, these tests have been performed now.

8. On page 4 it is incorrectly stated that TGFbeta ligates receptor tyrosine kinases instead of receptor serine/threonine kinases.

We apologise for this genuine error in our editorial efforts and thank the reviewer for their diligence in identifying its. It has been corrected.

Reviewers' comments:

Reviewer #1 (Remarks to the Author):

I have read over the revised manuscript and response to my previous comments. My concerns have been satisfied.

Reviewer #2 (Remarks to the Author):

Newman et al. (Wilkinson) Nature Communications Revised

This is a complex manuscript in which the authors propose a functional connection between autophagy, ATG5 and tumorigenesis in a cancer cell model cell line expressing mutant Ras, as well as a functional and molecular connection between ATG5 in autophagy, TRAF3, RELB and repression of TGF- β -induced gene expression through Smad complexes.

The authors provide a much improved manuscript, at least with respect to the weaknesses I found with the previous version. Too bad that the rebuttal was kind of argumentative, which made me at first rather apprehensive as to my evaluation of the revised manuscript.

In spite of the improvements, the major point of my critique remains, i.e. that Smad3 is generally seen as the major effector of TGF- β signaling, yet this was not and is now only barely taken into account. This relates primarily to the data in Fig. 7. Yes, the authors now show that RELB interacts with Smad3, but then focus on Smad4, which barely interacts with RELB, and the authors do not carry out any ChIP analyses for Smad3, and do not evaluate the effect of shRNA to Smad3 (and only evaluate Smad4 shRNA). Based on my substantial experience and knowledge, the model, as proposed, is wrong by not being willing to examine the role of Smad3. Based on Fig. 7A, RELB interacts strongly with Smad3 and only weakly with Smad4, and the latter interaction is most likely due to the presence of endogenous Smad3. At most genes, Smad3 is the TGF-activated Smad that directly interacts with DNA, and therefore it is most likely Smad3 that brings RELB to the DNA, using Smad4 as interacting partner. So, what can I say if the authors just do not want to take Smad3 into account? The problem is that one publishes a paper only once without later amendments and that the readers believe what is presented.

Related to this issue,

- Fig. 6B and 6C: Since this is a Smad3 reporter, the authors should test Smad3, Smad4 and Smad3+Smad4, rather than just resort to the previous data with Smad2 and Smad4, and then stating that they may act through endogenous Smad3 because it is a Smad3 reporter. I brought this up before, but this seems to be dismissed.
- page 13, lower paragraph: Fig. 7A does not "detect the formation of RELB complexes containing Smad4, Smad2 and Smad3" as concluded. One just cannot make that conclusion. Instead, Fig. 7A shows that RELB can interact with Smad2 and Smad3, and to a much lower level with Smad4, although, as mentioned, this may very well be through endogenous Smad3.

Additionally:

- In the Summary the authors still talk about Smad2/4 without taking into account Smad3.
- Page 4, line 5 from bottom: based on what we know, Smad2/3/4 complexes may be rather uncommon. Rather the heteromeric complexes generated in response to TGF- β are more commonly Smad3/Smad3/Smad4 and presumably also Smad2/Smad2/Smad4, although there is some evidence for genes responding to Smad2/3/4 complexes.

Reviewer #3 (Remarks to the Author):

The authors have addressed satisfactorily many of the points that were raised during the initial round of review. The finding that TRAF3 knockdown resulted in regain of growth of DeltaATG5A549 cells in vivo should be mentioned in the results. I still think that a second xenograft model with a different lung adenocarcinoma cell line would be essential to generalize the conclusions of the in vivo growth effects of TRAF3 upregulation upon autophagy inhibition.

We greatly appreciate the Reviewers' efforts in re-reading our revised manuscript. In response to points raised we respond as follows.

REVIEWER 2

- 1) The authors do not carry out any ChIP analyses for Smad3, and do not evaluate the effect of shRNA to Smad3 (and only evaluate Smad4 shRNA).**

We have now added data with Smad3 shRNA to test effect on RELB ChIP (see point 3 below).

We have also evaluated Smad2 shRNA for effects on RELB ChIP and Smad2 ChIP in existing Supp. Fig. S9C, where we obtained a similar pattern of results as with Smad4 shRNA.

- 2) Based on Fig. 7A, RELB interacts strongly with Smad3 and only weakly with Smad4, and the latter interaction is most likely due to the presence of endogenous Smad3.**

While we are glad that the reviewer sees evidence for interaction with all three Smad proteins, the affinity of RELB for Smad3 cannot be argued to be greater than that for Smad2 or Smad4. The fold enrichment for Smad3 immunoprecipitation is less impressive than for Smad2 and Smad4, given the baseline signal in the control (which is absent for Smad2 or Smad4). Notwithstanding, we make no other conclusions from this figure other than that RELB can exist in complexes with all three Smads (separate *or* multi-Smad isoform complexes, via direct *or* indirect binding to individual Smads).

- 3) At most genes, Smad3 is the TGF-activated Smad that directly interacts with DNA, and therefore it is most likely Smad3 that brings RELB to the DNA, using Smad4 as interacting partner.**

We have performed RELB ChIP after Smad3 knockdown and have not observed decreased RELB recruitment to promoters. In fact, an increase, albeit statistically insignificant over three biological replicates, was observed. So, while Smad3-driven transcription can be inhibited by RELB (see point 4 below), it appears the case that, at endogenous loci at least, Smad2 and Smad4 are primarily implicated in RELB recruitment. These new data are shown in Figure 7D and are discussed in changes to the text highlighted in pink on page 14.

- 4) Fig. 6B and 6C: Since this is a Smad3 reporter, the authors should test Smad3, Smad4 and Smad3+Smad4, rather than just resort to the previous data with Smad2 and Smad4, and then stating that they may act through endogenous Smad3 because it is a Smad3 reporter. I brought this up before, but this seems to be dismissed.**

Smad3 transfection is now shown to activate this reporter and this activity is repressed by RELB (new Figure 6C). Changes to the text are highlighted in yellow on page 13. So, RELB transfection can indeed also inhibit Smad3-driven transcription of this reporter.

Page 13, lower paragraph: Fig. 7A does not “detect the formation of RELB complexes containing Smad4, Smad2 and Smad3” as concluded. One just cannot make that conclusion. Instead, Fig. 7A shows that RELB can interact with Smad2 and Smad3, and to a much lower level with Smad4, although, as mentioned, this may very well be through endogenous Smad3.

This is a very good point, this sentence was ambiguous. Strictly speaking, Fig 7A shows that RELB can form complexes which contain Smad4, and complexes which contain Smad2, and complexes that contain Smad3, whether by direct or indirect binding to these Smads. It is possible that multi-Smad

isoform complexes form with RELB but our data not address this and we do not intend to claim this. Accordingly, we have clarified the text at Page 14, changes highlighted in yellow

Additionally:

- In the Summary the authors still talk about Smad2/4 without taking into account Smad3.

We agree wholly. We apologise for this oversight. We have corrected this now - changes are highlighted in green in the Summary.

- Smad2/3/4 complexes may be rather uncommon. Rather the heteromeric complexes generated in response to TGF- β are more commonly Smad3/Smad3/Smad4 and presumably also Smad2/Smad2/Smad4

We have elaborated upon this as requested, and/or clarified the text as required, highlighted in red (page 4).

REVIEWER 3

The finding that TRAF3 knockdown resulted in regain of growth of DeltaATG5A549 cells in vivo should be mentioned in the results.

This is now discussed at Page 11, highlighted in yellow and the data figures previously provided to the reviewer are now included in Supp Fig. 6D and E.

I still think that a second xenograft model with a different lung adenocarcinoma cell line would be essential to generalize the conclusions of the in vivo growth effects of TRAF3 upregulation upon autophagy inhibition.

The effects of autophagy on TRAF3 upregulation and TGF β gene transcription are shown to be active across three model systems.

In principle, the magnitude of consequent growth effects might indeed vary from one *in vivo* model to another – particularly given the context-dependent effects of the TGF β transcriptional program. Nonetheless, the model taken forward *in vivo* serves as a good proof-of-principle that the mechanism we describe for gene regulation downstream of autophagy has physiological output(s) in tumor cells, here in terms of growth.

Indeed, we do not wish to give readers the impression that this growth inhibition need always constitute the final output of regulation of TGF β by autophagy. So, we have re-read the text and made several modifications in the Summary and pages 5, 15 and 18 and in legend to Figure 9 on p55 (the model) to be clear on this point (highlighted in cyan).

REVIEWERS' COMMENTS:

Reviewer #2 (Remarks to the Author):

Newman et al. (Wilkinson) Nature Communications, Second Revision

This is a complex manuscript in which the authors propose a functional connection between autophagy, ATG5 and tumorigenesis in a cancer cell model cell line expressing mutant Ras, as well as a functional and molecular connection between ATG5 in autophagy, TRAF3, RELB and repression of TGF- β -induced gene expression through Smad complexes.

In the second revision, the authors now finally, reluctantly and minimally address the previous comments that stood out in my reviews of the first and second versions of the manuscript. This is now fine, although I regret that we had to go through this process of two revisions with its accompanying substantial effort, hand-holding and aggravation by this reviewer.

Some minor comments:

- Page 12, last sentence: I do not understand what you mean with "RELB fulfills the mechanistic criterion ... etc..." and frankly do not think that you can make this statement.
- page 13, line 4 from bottom, line 296: What do you mean with "abetted"?
- page 14, line 312: I do not agree that these experiments "demonstrate". They merely suggest. This sentence is confusing and should be simplified.
- The Discussion is unnecessarily long with many sentences and statements that are rather peripheral to the essence of the manuscript.
- The writing should be improved, as some statements are rather complex and/or unclear. A round of good editing should help.

Reviewer #3 (Remarks to the Author):

The authors have addressed adequately the issues that were raised during the previous round of review. I have no additional comments.

Response to Reviewer 2

We have modified the text of the manuscript to respond to comments on specific aspects of the writing:

- Page 12, last sentence: I do not understand what you mean with “RELB fulfills the mechanistic criterion ... etc...” and frankly do not think that you can make this statement.

We have simplified this sentence to merely say that RELB loss does not affect SMAD phosphorylation (line 273-275 in manuscript with tracked changes).

- page 13, line 4 from bottom, line 296: What do you mean with “abetted”?

We have changed “abetted” to “driven” in order to clarify this sentence (at line 297 in manuscript with tracked changes).

- page 14, line 312: I do not agree that these experiments “demonstrate”. They merely suggest. This sentence is confusing and should be simplified.

We have removed this sentence. The key point is actually made in the prior sentence, i.e. that SMAD(s) bind RELB. The nature of the experiment (coimmunoprecipitation from cellular lysates) dictates that there is no discrimination between direct or indirect binding. We don't see the need to belabour the point here (especially given the considerations below about the complexity of the manuscript).

See specific tracked changes at lines 312-315.

- The Discussion is unnecessarily long with many sentences and statements that are rather peripheral to the essence of the manuscript.

Many additions have been made to the Discussion in order to accommodate suggestions from prior rounds of Review. We appreciate that it has become unwieldy and inelegant. We have now edited and streamlined it, without losing the sense of the new points added in previous rounds of Review (see tracked changes in Discussion).

- The writing should be improved, as some statements are rather complex and/or unclear. A round of good editing should help.

Again, we have added to the manuscript substantially in the previous two rounds of review. We wholly agree it now needs trimming and have been back through it carefully, trying to simplify where possible (see tracked changes throughout).